# Using Fungi in Artificial Microbial Consortia to Solve Bioremediation Problems

**DOI:** 10.3390/microorganisms12030470

**Published:** 2024-02-26

**Authors:** Elena Efremenko, Nikolay Stepanov, Olga Senko, Aysel Aslanli, Olga Maslova, Ilya Lyagin

**Affiliations:** Faculty of Chemistry, Lomonosov Moscow State University, Lenin Hills 1/3, Moscow 119991, Russia; na.stepanov@gmail.com (N.S.);

**Keywords:** consortium composition, main trends, genetically modified cells, multicomponence effect, pollutant removal efficacy, immobilized forms

## Abstract

There is currently growing interest in the creation of artificial microbial consortia, especially in the field of developing and applying various bioremediation processes. Heavy metals, dyes, synthetic polymers (microplastics), pesticides, polycyclic aromatic hydrocarbons and pharmaceutical agents are among the pollutants that have been mainly targeted by bioremediation based on various consortia containing fungi (mycelial types and yeasts). Such consortia can be designed both for the treatment of soil and water. This review is aimed at analyzing the recent achievements in the research of the artificial microbial consortia that are useful for environmental and bioremediation technologies, where various fungal cells are applied. The main tendencies in the formation of certain microbial combinations, and preferences in their forms for usage (suspended or immobilized), are evaluated using current publications, and the place of genetically modified cells in artificial consortia with fungi is assessed. The effect of multicomponence of the artificial consortia containing various fungal cells is estimated, as well as the influence of this factor on the functioning efficiency of the consortia and the pollutant removal efficacy. The conclusions of the review can be useful for the development of new mixed microbial biocatalysts and eco-compatible remediation processes that implement fungal cells.

## 1. Introduction

Bioremediation processes are becoming increasingly important, both from the point of view of developing the scientific basis for their implementation, and the point of view of their practical application [1]. Soil and water ecosystems contaminated with oil and petroleum products [2], heavy metals [3], polycyclic aromatic compounds [4], and pesticides [5] have been mainly referenced in discussions of the objects of bioremediation. Now these pollutants [6] and microplastics [7] have been joined by pharmaceutical agents.

The biodegrading microorganisms of various pollutants have recently generated particular interest in studies of consortia of microorganism cells with heterogeneous composition, which combine the different metabolic potentials of participants [8,9,10]. In most cases, natural consortia contain a significant number of different prokaryotic cells (bacteria and archaea) that form biofilms, which may include aerobic and anaerobic cells [11]. At the same time, the extremely interesting properties possessed by those variants of consortia are created by cells of these prokaryotic microorganisms (i.e., bacteria and archaea) and eukaryotic microorganisms (in particular, filamentous fungi and yeast), which are able to coexist and catalyze biodegradation and bioremediation processes [12]. Such an artificial consortia, involving cells of microorganisms from different kingdoms [13,14,15,16,17], are especially attractive from a scientific and practical point of view, due to the possibility of expanding its total metabolic potential. At the same time, all the advantages that the artificial consortia of microorganisms themselves have over natural analogues, which have mainly been described in the literature on bacterial cells, are preserved:-easy-to-use reproducibility of the compositions of the consortia;-the possibility of introducing maximum targeted metabolic activity into the consortia cells, which are improved, including through the genetic modification of cells;-targeted variation of the ratios of cell concentrations in the consortium, to regulate the rates of associated biochemical processes catalyzed by cells;-sustainable functioning of the created consortia, which are accompanied by increased degrees of conversion of the original substrates, including toxic compounds [18,19,20,21,22,23,24,25,26].

The growing interest in fungi as part of artificial consortia is due to the fact that fungi are capable of synthesizing a very wide range of different hydrolytic and redox enzymes. A large number of pollutants can serve as substrates for these enzymes and undergo deep conversion due to their broad substrate specificity [12,17]. In addition, most of these enzymes are secreted and catalyze the destruction of substrates outside the cells of the fungi themselves, which provides more easily accessible sources of nutrition to all participants in the formed consortia. Moreover, fungi, being part of such consortia, are able to function in a wide range of pH and temperature values, in aerobic and anaerobic conditions [27,28,29].

In addition to the various catalytic capabilities that are characteristic of fungi, it has been noted that the morphology and biochemical composition of yeast cell walls, as well as the hyphae of filamentous fungi, can help reduce the level of toxicity of consortium habitats in the presence of various pollutants, including heavy metals, due to their biosorption [30].

Another mechanism for reducing the toxicity of media, which can be implemented by fungi in consortia with their participation in bioremediation processes, is based on their synthesis of lipopeptide biosurfactants. The appearance of these group of biosurfactants among metabolites can lead to biodeposition of pollutants [27]. In turn, the accumulation of organic acids, actively produced by fungi [31,32], as fungal metabolites in environments with functioning consortia of microorganisms promotes the formation of complexes with metals that prevent the migration of these pollutants.

Thus, the information that has been accumulated today indicates that the bioremediation of various environments, with the participation of fungal cultures as part of the used consortia, is implemented using biosorption, complexing and other physical and chemical processes, in addition to extracellular catalytic and intracellular metabolic reactions. Moreover, compared with the design of bacterial artificial consortia, in order to solve similar issues of bioremediation in synthetically formed biosystems containing fungal and yeast cultures, it is necessary to not only take into account the compatibility and antagonism of different microorganisms, but to also consider the features of cross-kingdom reactions [16,20]. In particular, chemically similar quorum-sensing signaling molecules such as lactone-containing molecules can appear in the microenvironment of the cells of different microorganisms (bacteria and fungi) which can, in turn, contribute to the formation of a synergistic effect from their participation in consortia.

On the other hand, it is known that fungi in a quorum state are capable of synthesizing antibiotics and mycotoxins, which limit their use in artificially formed consortia [33].

The purpose of this review is to analyze and compare the information accumulated in recent years on artificial consortia of different compositions with the obligatory participation of fungal cells in them. This analysis aimed to highlight the main priorities in modern experimental developments carried out by different researchers in the creation and application of mixed artificial consortia, and to also try to highlight current trends in the development of scientific and practical information in the field that is developing artificial mixed consortia with the participation of fungal cells.

## 2. Main Targets for Bioremediation Based on Various Consortia Containing Fungi

Recently, much attention has been paid to discussing the principles of designing artificial consortia [1,18,20,34,35,36]. In this review it was decided to analyze the success of already obtained consortia in the processes of bioremediation of various contaminated environments, taking into account the composition of microorganisms introduced into the consortia. In this regard, special attention was paid to exactly which microorganisms were used in consortia with fungal cultures to remove specific pollutants (heavy metals, dyes, synthetic polymers, pesticides, polycyclic aromatic hydrocarbons, pharmaceutical agents, crude oil, phenols, etc.), and to the degree that their removal was achieved.

### 2.1. Removal of Heavy Metals

Heavy metals which, as a result of anthropogenic pollution, enter the environment and accumulate in food chains, have a negative effect on the state of the cells of various living organisms [37]. Lead, cadmium, and chromium are highly toxic, mutagenic, and carcinogenic, and the presence of several metals in contaminated objects at once can aggravate their toxicity. Metal ions can interact with hydroxyl, carboxyl, phosphate and sulfhydryl groups, causing changes in the properties of proteins, nucleotides, coenzymes, phospholipids and other substances involved in cell metabolism. In addition, metal ions can be replaced in the active centers of enzymes, causing inhibition of their activity [1].

Soil contamination with heavy metals leads to a decrease in the species diversity of microbial communities, although the cells of individual microorganisms are capable of accumulating heavy metals, due to their sorption and intracellular complex formation. The most active participants in these processes are polysaccharides of the cell walls of various microorganisms [38], especially those that are part of consortia (Table 1) [39,40,41,42,43,44].

Bioremediation of heavy metals by microorganisms includes the following main mechanisms: adsorption, bioaccumulation, biotransformation and bioleaching [45]. Microbial consortia of different compositions are characterized by different abilities to remove heavy metals. The majority of interest is focused on those options that can be used for bioremediation of objects contaminated simultaneously with several heavy metals. For example, a fungal consortium consisting of several species of fungi of the genus *Aspergillus* was used to remove heavy metals [39].

It was found that such a consortium leads to an increase in the removal of heavy metals Cr, Zn, Pb, Cd and Ni by 2–45%, in comparison to individual fungal cultures included in the consortium. At the same time, the efficiency of bioremediation of heavy metals under the influence of such a consortium reached 70–90%. Interestingly, Cr removal was maximum in the consortium of *A. niveus* and *A. flavus* (93%), while in the combination of *A. niger* with *A. flavus* or *A. niveus* cells it decreased by 3%. Thus, the results obtained from combining fungi belonging to the same genus were slightly different.

The genetic proximity of fungi used in consortia does not lead to a significant improvement in their characteristics in bioremediation processes [40]. Thus, an attempt by other researchers to increase the efficiency of removal of Cr(VI)) and Cd(II) (100 mg/L) by a fungal consortium consisting of a strain of *A. flavus* and two closely related strains of *A. fumigatus* only made it possible to obtain 80% for each of the metals, and slightly more than 70% when processing a mixture of both metals by this consortium. 

A fungal consortium consisting of 13 strains of various fungi (*Perenniporia subtephropora*, *Aspergillus fumigatus*, *A. niger*, *Phanerochaete concrescens*, *Cerrena aurantiopora*, *Polyporales* sp., *Fusarium equiseti*, *F. chlamydosporum*, *Paecilomyces lilacinus*, *Tremates versicolor*, *Antrodia serialis*, *Daldinia starbaeckii*, *Penicillium cataractum*) was effective in removing metals As (77%), Mn (71%), Cr (60%), Cu (52%) and Fe (52%) from the soil within 100 days [41].

It was observed that when treated with contaminated soil the number of fungal cells increased until day 60, and then decreased. Most likely, this could be due to cells accumulating a certain maximum concentration of metals, which then began to have a negative effect on the functioning of the consortium. In support of this explanation, when the same consortium was used in another study to remove Ni, Pb, Zn from soil, the metal removal efficiency was 32–52%, as a result of bioaccumulation [42].

In optimal conditions (in the presence of glucose and peptone, pH 5.0, 30 °C, 120 h), the efficiency of the microbial consortium consisting of *A. fumigatus*, *A. terreus* and *Paenibacillus dendritiformis* in removing Cd was 95% [43]. Thus, in this case, adding just one “outsider” (that is, bacteria *P. dendritiformis*) to the consortium of filamentous fungi belonging to the same genus led to a more significant efficiency of the process than when using only “close relatives” in the created consortium.

In a similar situation with two representatives of the genus *Aspergillus* (*Aspergillus terreus*, *A. flavus*), the introduction into a consortium of not one, but two, “outsiders” (*Talaromyces islandicus*, *Neurospora crassa*) for the removal of Pb, Ni from real wastewater gave a guaranteed high efficiency of simultaneous removal of both metals (95–97%) [44].

In general, known examples of the use of fungal consortia clearly show the successful removal of heavy metals through their adsorption and accumulation in cell biomass with higher efficiency, compared to the use of individual strains of microorganisms. At the same time, the most attractive option is to combine a small number of participants in such consortia, for instance combining fungi from the same genus with the addition of “exogenous” representatives, presumably so they are playing the role of competitors or antagonists.

### 2.2. Decolorization of Dyes

Various industries use dyes, including synthetic ones, with complex aromatic molecular structures, whose degradation poses a serious problem [46]. Many physicochemical processes, including adsorption, chemical oxidation, and photocatalysis, are effective in decolorizing wastewater containing dyes. However, high cost and the formation of toxic secondary compounds limit the use of these treatment methods. For comparison, the biological degradation of a number of dyes, for which the use of fungi has been shown to be highly effective, is relatively more economically profitable and environmentally friendly [47].

Decolorization and degradation of dyes by fungi are one of the most studied processes, accounting for hundreds of articles published in the field [48]. Fungi remove dyes through processes such as biosorption, biodegradation, and bioaccumulation. The degradation potential depends primarily on fungal metabolism and the expression of extracellular enzymes such as lignin peroxidase (LiP), manganese peroxidase (MnP) and laccase. However, high efficiency has only been established for the decomposition of certain dyes, while wastewater from enterprises is usually a mixture of various organic and inorganic pollutants, including dyes. The use of microbial consortia may therefore be more effective for dye removal (Table 2) [49,50,51,52,53,54,55,56].

Yeasts, as eukaryotic and single-celled fungi, also have practical applications in wastewater treatment, due to their high degradation activity in various azo dyes. In particular, a yeast consortium consisting of three different cultures was obtained that was capable of biodegrading industrial dyes [49].

It was shown that when such a yeast consortium is used, various dyes (100 mg/L) can be degraded within 6–12 h with an efficiency of 90–100%. In real wastewater from the textile industry, a significant reduction in the content of total organic carbon (54%), biological oxygen demand (74%) and chemical oxygen demand (95%) and color removal (98%) was shown within 24 h at 30 °C and pH 8.0 in the same consortium. The synthesis and secretion by cells of the consortium of a number of redox enzymes (laccase, veratryl alcohol oxidase, tyrosinase, azoreductase, NADH-DCIP reductase), that are found in the reaction medium, ensured the efficiency of the process [49].

The efficiency of bleaching a dye such as cibacron brilliant red 3B-A by a consortium of fungi *Daldinia concentrica* and *Xylaria polymorpha* reached 98–99%, and was higher in comparison with the effect of the same cultures used separately [50]. Among the enzymes involved in dye degradation, laccase, lignin peroxidase and Mn-containing peroxidases were identified. The products of dye metabolism exhibited low toxicity.

A synergistic interaction has been shown between cells of the yeast *Rhodotorula* sp., in a consortium with the bacteria *Raoultella planticola* and *Staphylococcus xylosus*, in the process of biodegradation of methylene blue [51]. Intracellular NADH and DCIP reductases were the key enzymes involved in the process of dye degradation. In addition, azoreductase, tyrosinase, laccase, nitrate reductase, MnP and LiP also contributed significantly to the degradation process of methylene blue. The authors of the study confirmed the phytotoxicity and cytotoxicity of dye degradation by-products. When cells of the same consortium immobilized in a Ca-alginate gel were used to purify municipal wastewater and industrial effluents contaminated with the same dye (200 mg/L), the degradation efficiency of methylene blue reached 100% and 78.5%, respectively, when treatment was carried out within 144 h.

A mixture of 15 azo dyes (20 mg/L of each type) was subjected to microbial degradation under the influence of fungal (*A. niger*, *A. terrus*, *A. oryzae* and *A. fumigatus*) and bacterial (*Brevibacillus brevis*, *Bacillus coagulans*, *Lysinibacillus macroides*, L. *fusiformis* and *B. subtilis*) consortia [52]. The most active bacterial strain was *B. subtilis*, which showed the highest degree of biodegradation (from 71.8% to 100%) of eight azo dyes. However, under the influence of the bacterial consortium, the degradation efficiency was lower than that of the fungal consortium, which turned out to be capable of biodegradation of all 15 tested azo dyes. At the same time, 10 azo dyes were degraded by 100% under the action of the fungal consortium.

The anthraquinone dye Disperse Red 3B (0.1 g/L) was decomposed by a consortium consisting of cells of the fungus *Aspergillus* sp. and microalgae *Chlorella sorokiniana* for 4 days, with an efficiency of 98.1% [53]. In this case, the resulting low-molecular-weight dye destruction products were characterized by low toxicity. When the dye concentration increased by five times, the bleaching efficiency decreased to 69.3%, while when consortium cells were used separately, the bleaching efficiency was almost seven times lower (less than 10%).

The efficiency of methyl orange bleaching by mixed cultures of the fungus *Daedalea dickinsii* and bacteria *P. aeruginosa* reached 98% within 7 days [54], while the efficiency of dye degradation by one fungus was only 67.5%.

To degrade azo dyes, a yeast consortium consisting of two cultures (*Sterigmatomyces halophilus* and *Meyerozyma guilliermondii*) was obtained [55]. In optimal conditions (pH 7.0, 35 °C and 50 mg/L dyes), the efficiency of degradation of dyes (Reactive Black 5, Acid Orange 7; Reactive Green 19, Reactive Yellow, ABC, Atlantic Black C) was 88–97%. Moreover, NADH--DCIP-reductase and azoreductase were the main enzymes involved in this process.

A consortium consisting of two filamentous fungi (*Penicillium oxalicum* and *Aspergillus tubeingensis*) provided effective degradation (96%) of Congo red within 12 h, which was significantly better when compared to the same individual strains (30–69%) [56]. Presumably, the increased degradation could be the result of mutualism between the two cultures. The products obtained as a result of the destruction of Congo red were non-toxic for the growth of microorganisms and plants. An increase in the degree of dye degradation positively correlated with the production of laccase and Mn-containing peroxidase by fungi.

Thus, when using microbial consortia containing fungi or yeast, it is possible to effectively decompose dyes, including those present in the mixture in real wastewater. The main role (in such bioremediation of water sources contaminated with various dyes) is played by complexes of oxidoreductases secreted by fungal cells.

### 2.3. Destruction of Synthetic Polymers

Synthetic polymers, also known as man-made polymers, are polymers that are artificially produced by humans. Synthetic polymers can be classified into four distinct categories, namely thermoplastics, thermosets, elastomers and synthetic fibers. Synthetic polymers are derived from petroleum oil. The most common types of synthetic polymers are: polyethylene, polypropylene, polyvinyl chloride, polystyrene, polyurethane, polyvinyl chloride, polyethylene terephthalate [7].

The use and production of polymer materials is constantly increasing, and the amount of waste that enters the environment is increasing at the same time [57]. During the process of degradation, the size of plastic particles decreases, increasing the possibility of them penetrating into the cells of microorganisms, animals and plants, thus creating their potential toxicity. In addition, heavy metals, polycyclic aromatic hydrocarbons, pesticides, pharmaceutical compounds and other pollutants can be adsorbed onto plastic microparticles [58]. Microplastics are tiny plastic particles less than 5 mm in size [7]. It has been found that microplastic particles can change the formation of soil aggregates and the bulk density and water-holding capacity of the soil, which ultimately leads to changes in the composition of microbial communities and a decrease in plant growth and development [59].

Today, many works study the destruction of microplastics by using physicochemical (thermoconversion, hydrogenolysis, silylation, electrooxidation and photolysis, solvolysis) and biocatalytic (enzymatic hydrolysis) methods [7]. However, physicochemical processes are expensive, and require high temperatures and energy, and need special equipment that uses strong acid or metal catalysts; in addition, the range of microplastics effectively decomposed by individual enzymes is limited to polyesters and polyamides [7]. As a result, microbial degradation of polymers remains a relevant preoccupation for researchers, with artificial consortia generating maximum interest (Table 3) [60,61,62,63,64,65,66,67].

Biodegradation of microplastics is analyzed, above all, by determining the weight loss of a polymer sample, changes in its mechanical properties, and/or the appearance of destruction products. More than 90 microorganisms, including fungi, have been identified that are capable of degrading synthetic polymers [68,69]. It should be noted that biodegradation is a long-term process. It is possible to intensify it through pretreatment of the original polymers that uses chemical or physical methods [70].

More efficient degradation of plastics can only be achieved by the action of microbial consortia, since complexes of enzymes that cannot be synthesized by a single microorganism are required. For example, to achieve the degradation of low-density polyethylene (LDPE), a consortium of three yeast strains *Sterigmatomyces halophilus*, *Meyerozyma guilliermondii* and *M. caribbica* was obtained [60], which provided a decrease in the strength of the polymer and weight loss by 63.4% and 33.2%, respectively, within 45 days The yeast were capable of synthesizing LDPE-degrading enzymes (such as Mn peroxidase, laccase, and lignin peroxidase), whose activity in the consortium increased. The development of the biodegradation process was accompanied by the active accumulation of the biomass of yeast destructor cells, and the products of polymer metabolism were alkanes, alkenes, carboxylic acids, aldehydes, ethers and alcohols.

The process of biodegradation of polyurethane by a consortium consisting of cells of the fungus *A. niger* and bacteria *P. aeruginosa* was studied [61]. Cellulose (4 wt%) was added to the medium to stimulate the biodegradation of polyurethane by intensifying the induced synthesis of the necessary enzymes and, as a result, the weight loss of polymer was about 20%.

A fungal consortium consisting of strains of *Curvularia lunata*, *Alternaria alternata*, *Penicillium simplicissimum* and *Fusarium* sp. degraded polyethylene, producing 27% weight loss within 3 months. Degradation by individual strains of polyethylene, meanwhile, was low and amounted to only 0.7–7.7% [62]. Similar results were obtained during the biodegradation of LDPE by a consortium consisting of three strains of fungi of the genus *Aspergillus* (*A. niger*, *A. flavus* and *A. oryzae*), with weight loss of polymer about 26% during 55 days [63].

Consortia of microorganisms obtained by isolating the most active strains from various sources (activated sludge, compost and river bottom sediments) were used to compile consortia and degrade polymeric materials [64]. The isolated microorganisms were bacteria of the genus *Bacillus* and *Pseudomonas*, fungi of the genera *Aspergillus*, *Rhizopus*, *Alternaria*, *Penicillium* and *Trichoderma*, as well as the yeast *Candida parapsilosis*. A polymer sample consisting of LDPE, thermoplastic starch and styrene-ethylene-styrene was biodegraded by the resulting consortia, with 16–22% weight loss within 56 days.

Microbial consortia, composed of microorganisms isolated from enriched landfill soil, were used to degrade LDPE for 90 days, resulting in polymer weight reduction to 55.6% [65]. The authors attribute the reason for such a long process, with a relatively low percentage of polymer destruction, to the large composition of the consortium.

The effect of extrusion cycles on the degradation of a polypropylene/poly(butylene adipate-*co*-terephthalate)/thermoplastic starch mixture by a fungal consortium consisting of only two filamentous fungi (*Aspergillus* sp. and *Penicillium* sp.) was studied for 30 days [66]. Extrusion for seven cycles increased the biodegradation efficiency by two times. However, the weight loss of the polymer was still insignificant (2.3%).

Individual microorganisms (bacteria *Bacillus* sp. and fungi *Aspergillus* sp.) were introduced into a consortium degraded LDPE with an efficiency of 10%, while a microbial consortium consisting of the same bacteria and fungi showed slightly more efficient degradation of the same polymer as a whole (12%). However, no antagonistic effect was observed between the cells of the consortium [67].

In general, it is worth noting that the biological degradation of synthetic polymers is still a long process, despite the constant search for new solutions in this area, including the use of new microbial consortia. In this regard, it may be more effective to combine physicochemical and biological methods with the use of microbial and enzymatic catalysts.

### 2.4. Degradation of Pesticides

Pesticides are widely used in agriculture to control insects, weeds, pathogenic fungi and bacteria. Excessive and uncontrolled use of pesticides leads to serious environmental problems, including their accumulation in soil and water sources [71]. In some cases, even after wastewater treatment at the treatment plants, the concentrations of pesticides can remain quite high [72]. The use of biological methods is considered to be an effective approach to the degradation of pesticides, but the efficiency of such processes may be low and they may have a significant duration [73]. For this reason, the search for new strategies for the biocatalytic elimination of pesticides continues to be relevant, and the use of microbial consortia, including those containing fungi, seems to offer a promising solution to this problem (Table 4 [74,75,76,77,78,79]).

The use of a consortium of the bacteria *Bacillus subtilis* and the fungus *Fomitopsis pinicola* to achieve the biodegradation of the pesticide DDT (1,1,1-trichloro-2,2-bis(4-chlorophenyl) ethane) is known. Within 7 days at 30 °C, the degree of pesticide degradation reached 86% [74]. Similar results for the degradation of DDT, when the efficiency of the process was 86%, were obtained using a consortium consisting of the fungus *Pleurotus ostreatus* and the bacteria *Pseudomonas aeruginosa* [75].

A consortium of cells from the microalgae *Chlorella vulgaris* and the fungus *A. niger* was used to remove pesticides from water. When treating a mixture of 38 pesticides at a total concentration of 72.7 µg/L, the biodegradation efficiency reached 23%. Moreover, the final concentrations of some pesticides (difenoconazole, carfentrazone ethyl, phenmedipham and trinexapac ethyl) were, after treatment by the consortium, even below the detection limit [76].

The efficiency of biodegradation of pesticides (atrazine, iprodione, chlorpyrifos), under the action of a suspended fungal consortium consisting of *Verticilium* sp. cells and *Metacordyceps* sp. [77], reached 81–99% within 21 days. Laccase, MnP, and Mn-independent peroxidase were the main enzymes involved in pesticide degradation. The use of the same consortium in an immobilized form (incorporation into a Ca-alginate gel) for the degradation of pesticides, in a packed-bed bioreactor in a continuous mode, ensured that, at a medium flow rate of 90 mL/h, 64–96% of pesticide degradation was achieved. The performance of the process depended on many factors, including the flow rate of the medium, the type of pesticide itself and the time of its processing.

The efficiency of biodegradation of pesticides (atrazine, carbendazim, carbofuran, metalaxyl) by a microbial consortium (composed of microorganisms present in coconut fiber, garden compost and agricultural soil) supplemented with the fungus *Trametes versicolor* was assessed [78]. This consortium effectively biodegraded all tested pesticides by 72–99% within 16 days. The addition of the antibiotic oxytetracycline (10 mg/kg) did not significantly affect the efficiency of pesticide degradation by this consortium.

A synergistic effect was observed during the degradation of DDT by a mixed consortium composed of the fungi *Fomitopsis pinicola* and bacteria *Ralstonia pickettii* [79]. The degradation of the pesticide after 7 days was 61%; when the same microorganisms was used separately; the degradation of DDT was 31–42%. Degradation of DDT occurred through its transformation into 1,1-dichloro-2,2-bis(4-chlorophenyl)ethane, which was further transformed into 1,1-dichloro-2,2-bis(4-chlorophenyl)ethylene, and eventually converted to 1-chloro-2,2-bis(4-chlorophenyl) ethylene. These metabolites were less toxic than DDT. Fungi *F. pinicola* produced extracellular enzymes, such as oxidoreductase, superoxide dismutase, catalase, laccase and cytochrome P450 monooxygenase. Cells of the bacteria *R. pickettii* synthesized monooxygenase, lipase, and depolymerase. *F. pinicola* and *R. pickettii*, which were shown to be in co-metabolism, since *F. pinicola* fungi converted DDT into products that were used by bacterial cells. When fungal cells grew together with bacteria, the formation of a thick layer of hyphae was observed, indicating that the bacterial cells stimulated mycelial growth.

In general, the presence of fungi in mixed artificial consortia makes it possible to more effectively catalyze the decomposition of pesticides (both individually and in mixtures), in comparison with individual cultures. However, concerns arise about the possible formation of resistant strains meaning that, along with the creation of consortia, it is necessary to develop methods for regulating their composition and metabolic activity.

### 2.5. Degradation of Polycyclic Aromatic Hydrocarbons

Polycyclic aromatic hydrocarbons (PAH), formed as a result of human anthropogenic activities, constitute a major class of environmental pollutants, some of which are toxic and resistant to degradation [80].

Along with PAH, other contaminants (heavy metals, pesticides and other pollutants) may be simultaneously present, which can lead to an increase in the overall toxicity of the media and, most importantly, complicate bioremediation. Various physicochemical methods for the remediation of PAHs are currently available, but they have several disadvantages, including high cost, difficulty in implementation, and, sometimes, inefficiency [81].

Various individual strains of microorganisms, including microalgae, fungi and bacteria, have demonstrated the ability to biodegrade PAHs [82]. Bacterial destruction of PAHs consists of their enzymatic transformation under the action of monooxygenases or dioxygenases with the formation of intermediate compounds such as catechols. Some fungi of the genera *Cunninghamella*, *Coriolopsis*, *Chrysosporium*, *Trichoderma*, *Phanerochaete*, *Aspergillus*, *Pleurotus*, *Trametes*, *Bjerkandera*, *Penicillium*, *Mucor* and *Cladosporium* can metabolize PAHs as a sole carbon source or co-metabolize them when using other substrates.

Microalgae have various mechanisms for the decomposition or removal of PAHs, namely biosorption, bioaccumulation, biodegradation, and complexation [83]. The biodegradation of PAHs by ligninolytic and non-ligninolytic fungi is characterized by two different main mechanisms that occur due to various enzymes mainly involved in these processes. Secreted laccase, MnP, and lignin peroxidases of ligninolytic fungi are the most active catalysts acting on the aromatic rings of PAHs [84]. Oxidases can participate in the formation of free hydroxyl radicals by giving up one electron, which oxidizes PAH rings outside fungal cells. The products of such reactions are PAH-quinones and acids, whereas intracellular cytochrome P450 monooxygenases, and hydroxylating low molecular weight PAHs, are the main participants in the oxidative degradation of PAHs carried out by non-ligninolytic fungi [84]. Cytochrome P450 monooxygenases oxidize PAHs to epoxides and dihydrodiols. However, in artificial microbial consortia of mixed composition, a synergistic metabolic effect can develop, leading to improved degradation of PAHs by consolidating various catalytic mechanisms (Table 5 [85,86,87,88,89,90,91,92]). As has been noted [81], PAHs are often present in the environment as a mixture, and bioremediation can therefore occur through co-metabolism. For example, for the biodegradation of PAHs in the presence of heavy metals, a microbial consortium consisting of *B. subtilis* and the fungus *Acremonium* sp. was used, which effectively decomposed (61–100%) naphthalene, fluorine, phenanthrene, anthracene and fluoranthene within 10 days [85].

The study of the biodegradation of benzo[a]pyrene by cells of the fungus *Pleurotus ostreatus*, as well as consortia of this fungus with *Penicillium chrysogenum* or with the bacteria *P. aeruginosa* is known. The efficiency of benzo[a]pyrene degradation by consortia (86.1 and 75.1%, respectively) was higher than that of an individual fungal strain (64%) [86].

A mixed consortium was created, in which cells were immobilized on biochar to degrade phenanthrene (50 mg/L) while removing 150 mg/L Cd^2+^. The dominant species were bacteria *Proteobacteria* and *Bacteroidota*, and fungi of *Fusarium* genus [87]. The immobilized consortium effectively decomposed phenanthrene (up to 98%) within 7 days, while simultaneously removing up to 99% of Cd^2+^. Immobilization of the consortium improved its dehydrogenase activity, which increased the degradation of phenanthrene. In addition, immobilization led to stabilization of the composition and more favorable development of the microbial community in the consortium.

Using the method of induced microbial selection, microbial consortia were created, including natural fungal (*Aspergillus flavus*, *A. nomius*, *Rhizomucor variabilis*, *Trichoderma asperellum*, and *A. fumigatus*) and bacterial strains (*Klebsiella pneumoniae*, *Enterobacter* sp., *Bacillus cereus*, *Pseudomonas aeruginosa*, *Streptomyces* sp., *Klebsiella* sp., and *Stenotrophomonas maltophilia*), as well as genetically engineered strains of *A. niger* expressing lignin peroxidase and MnP genes for the degradation of phenanthrene, pyrene and benzo[a]pyrene in soil [88]. The efficiency of degradation of a mixture of PAHs present in the soil at a concentration of 1 g/kg under the influence of this consortium reached 65–92% after 14 days of the process.

Cells of a microbial consortium consisting of the bacteria *Kocuria rosea* and the fungus *A. sydowii* were used after immobilization in guargum-nanobentonite composite water dispersible granules to degrade a mixture of PAHs (naphthalene, fluorene, phenanthrene, anthracene, and pyrene) containing 100 μg/g of each pollutant in sandy loam soil. Cell immobilization ensured PAH degradation by 85–100% [89].

An increase in the efficiency of pyrene degradation (up to 78%) was noted when using a microbial consortium consisting of bacteria *P. putida* and yeast *B. persicus*, when rhamnolipid was added to the medium as a biosurfactant. The use of rhamnolipid led to an increase in the bioavailability of pyrene [90].

A microbial consortium including *Ochrobactrum intermedium* and *Pleurotus ostreatus* was able to degrade a mixture of PAHs in soil [91]. The consortium completely removed fluoranthene, indene[1,2,3-cd]pyrene and benzo[g,h,i]perylene within 50, 80 and 50 days, respectively. Anthracene, pyrene, chrysene and benzo[a]anthracene were degraded by 86–98% in 110 days. The consortium was more effective than the two microorganisms used alone. Bacteria formed biofilms around fungal hyphae, stabilizing the high functional activity of the consortium.

The degradation of PAHs (anthracene, phenanthrene, fluorene, pyrene, and fluoranthen) was studied using a mixed consortium containing *P. ostreatus* and *A. brasilense* [92]. Compared to monocultures (20–60%), the efficiency of PAH degradation by the consortium exceeded 70%, due to an increase in the synthesis of extracellular laccase and versatile peroxidase. At the same time, an increase in the mycelium biomass and the number of bacterial cells was found in the consortium, which confirmed the utilization of PAH destruction products.

In analyzing the presented data, one can note the high efficiency of mixed consortia in biocatalytic processes of PAH decomposition. It turned out that it was possible to increase the efficiency of consortia, including through the use of genetically improved strains [88]. However, despite the efficient functioning of recombinant cells, their use is associated with a number of problems, with the need to maintain selective cultivation conditions for them presenting itself as a particular problem. It is also possible to increase the efficiency of degradation by using immobilized forms of consortia [87,89]. In this case, it is advisable to use the methods of incorporating cells into gel matrices and sorption of cells on various carriers as the main methods of immobilization.

### 2.6. Degradation of Pharmaceutical Pollutants

The active use of various pharmaceutical compounds (antibiotics, hormones, analgesics, anticonvulsants, anti-inflammatory, cardiovascular, antiepileptic and other drugs) has led to them contaminating environmental objects. The presence of these substances in wastewater is detected in concentrations from 0.15 ng/L to 2.0 g/L [93]. These compounds are most often highly toxic and resistant to decomposition, and can therefore accumulate, causing harm to microorganisms and various animals.

Physicochemical processes (membrane filtration, ozonation, and photocatalytic oxidation, sorption, etc.) do not provide complete removal of pharmaceutically active compounds (PhACs) and are characterized by a fairly high cost, the formation of toxic by-products, low selectivity, etc. [6].

The biocatalytic transformation of PhACs under the action of enzymes such as laccase and peroxidases has become widespread. However, a number of problems limit their use including, in particular, the low activity of laccases at neutral pH values or the need to add H_2_O_2_ as a substrate for peroxidases. In addition, it is necessary to ensure the effective and long-term functioning of enzymes, which is achieved by immobilizing them and increasing the cost of the resulting enzyme preparations [6].

It has also been established that the efficiency of enzymatic hydrolysis sharply decreases when treating real wastewater, which is because the micropollutants and solid suspended microparticles can significantly reduce the activity of enzymes, acting as their inhibitors and sorbents [6]. In addition, enzyme complexes are needed. In this regard, microbiological wastewater treatment is still relevant [94], and the use of artificial consortia for the bioremediation of PhACs seems to be the most effective approach, as this ensures the simultaneous synthesis by different cells of microorganisms of different enzymes involved in the destruction of pollutants (Table 6 [95,96,97,98,99,100,101]). In particular, in an artificial consortium of fungi consisting of *Phanerochaete chrysosporium* and *Pycnoporus sanguineus*, 100% degradation of ciprofloxacin, norfloxacin and sulfamethoxazole (SMX) was shown, due to the presence of a complex of oxidative enzymes (laccase, lignin peroxidase and Mn peroxidase) [95]. When a pure culture of *P. chrysosporium* was used, the efficiency of antibiotic degradation was lower (64.5%, 73.2% and 63.3%, respectively), and the adsorption of antibiotics by fungal mycelium in the consortium was only 2–3%.

SMX has been shown to significantly inhibit the laccase activity of the fungus *P. sanguineus*, which reduces the biodegradation efficiency of this antibiotic. To eliminate this problem, a consortium of this fungus with bacterial cells of *Alcaligenes faecalis* was used [96]. The removal efficiency of SMX within 24 h by the consortium in a medium containing a mixture of vitamins (VB2, VB6, VB12, and VC) was higher (93%) than when only pure fungal culture was used (28%). It was confirmed that the fungal mycelium adsorbs only 1% SMX—that is, the reduction in antibiotic concentration was achieved precisely due to biotransformation.

Biodegradation of a mixture of nonsteroidal anti-inflammatory drugs (celecoxib, diclofenac and ibuprofen) was carried out under the influence of two fungal strains *Ganoderma applanatum* and *Laetiporus sulfureus* for 72 h [97]. The efficiency of pollutant removal by the consortium was 99.5%, while the same microorganisms individually provided 61 and 73% degradation, respectively. A significantly increased induction of synthesis of a number of enzymes (laccase, lignin peroxidase and MnP) in the consortium was revealed (by 201, 180 and 135%, respectively).

Degradation of carbamazepine, diclofenac and ibuprofen (1 mg/L) was carried out by a multicomponent fungal consortium (*A. niger*, *Mucor circinelloides*, *Trichoderma longibrachiatum*, *Trametes polyzona* and *Rhizopus microsporus*) in a sequencing batch reactor [98]. It was shown that the enzymatic activity of laccase, MnP and lignin peroxidase increases with increased biomass. The efficiency of pollutant removal within 7 days was more than 90% (Table 6).

Water purification from seven pharmaceutical drugs (acetaminophen, carbamazepine, diclofenac, metoprolol, naproxen, ranitidine and SMX) was carried out using cells of microalgae *C. vulgaris*, the fungus *A. niger* and their consortium. After 72 h, and only when the consortium was used, a decrease in the initial concentration of ranitidine by 50% was noted. For other substances, the purification efficiency was insignificant, which was due to the low concentration of the used consortium in the reaction medium (it was 16–500 times lower than what is usually used in wastewater treatment plants) [99].

Various microorganisms (fungi and bacteria) isolated from sewage sludge were studied to obtain artificial microbial consortia capable of effectively degrading pharmaceutical pollutants (diclofenac, carbamazepine and ketoprofen). It was shown that consortia with the fungi *Penicillium oxalicum* and *Penicillium rastrickii* are able to effectively degrade diclofenac (99%) and ketoprofen (80%) within 10 days. At the same time, under the influence of individual crops, the efficiency of degradation was low [100]. It is important to note here that while the consortia showed high efficiency, their ability to degrade pollutants could be suppressed by the appearance of “negatively” acting species of microorganisms, leading to the cessation of interactions between the components of the consortia and their trophic connections. Thus, such a “negative” action of microorganisms on the consortium at a certain stage can be considered as a regulator of its bioremediation activity that allows it to slow down the speed of functioning of such artificial cellular biosystems, if necessary.

A consortium based on the microalgae *C. vulgaris* and the fungi *A. oryzae* was used for the biodegradation of a mixture of antibiotics (sulfamonomethoxine, SMX and sulfamethazine) [101]. At the same time, different concentrations of copper (Cu(II)) were introduced into the medium. The results confirmed that the addition of Cu(II) (0.1–0.5 mg/L) increased the removal of sulfamonomethoxine (58.8%), SMX (63.5%) and sulfamethazine (63.9%). The degradation pathways of these compounds were associated with hydroxylation, deamination, oxidation, desulfurization, and the breaking of chemical bonds. The fungal mycelium sorptionally retained microalgae cells on its surface, helping to stabilize the consortium. Copper ions had a great influence on the accumulation of microalgae biomass, since they can absorb and use Cu(II) in photosynthesis (instead of magnesium) in the chlorophyll structure, continuing to fix CO_2_ and synthesize carbohydrates. In addition, it was found that Cu(II) increased the lipid content of *Chlorella* cells in the co-culture system.

These examples indicate the high efficiency (50–100%) of the use of artificial microbial consortia with fungal cells in the processes of bioremediation of wastewater contaminated with pharmaceutical pollutants. At the same time, mainly fungal oxidoreductases are involved in the implementation of the decomposition mechanisms of pollutants.

### 2.7. Elimination of Pollutant Mixtures

Today, pollution of various environmental objects with various chemical substances (natural and synthetic) can have a negative impact on living objects, which is of particular concern as it raises questions about the need for their analytical control and subsequent elimination. Here, one of the most effective solutions to the problem seems to be the use of mixed artificial consortia that include cells of different fungi in their composition (Table 7 [102,103,104,105,106,107,108,109,110,111]).

To clean the soil contaminated with crude oil, a consortium consisting of cells of the bacteria *Acinetobacter baumannii* was combined with cells of the fungi *Talaromyces* sp. andused. The degree of decomposition of oil was 65.6% after soil treatment for 28 days [102]. In another similar work, not one, but two, bacterial strains (*Paraburkholderia* sp. and *Paraburkholderia tropica*) were combined with the fungi *Scedosporium boydii* into an artificial consortium, which was also used for biodegradation of crude oil pollution [103]. Within 1 week, the efficiency of biodegradation reached 81.4% with a mixed consortium. This parameter was 20% higher, compared to the combined use of two types of bacterial cells. At the same time, the formation of bacterial biofilms on the surface of the hyphae of the fungus was revealed, which stabilized the mixed consortium.

Cells of the fungi *Scedosporium* sp. were combined with bacteria *Acinetobacter* sp., producing biosurfactants for the decomposition of crude oil. The results showed that the efficiency of oil degradation (200 mg/L) increased from typical for individual microorganisms, increasing 23–29% to 58.6% within 7 days of an artificial consortium being used [104]. The ability of bacteria to produce biosurfactants leads to an increase in the effective biodegradation of hydrophobic substrates.

Greywater refers to domestic wastewater generated after sewage and washing machine effluents, which contain various pollutants, including detergents, dyes, heavy metals and personal care products. A bacterial-fungal consortium with *Micrococcus luteus*, *Rhodococcus equi* and *A. niger* was used to purify greywater [105]. In optimal operating conditions (pH 7, 35 °C), the decrease in the content of pollutants (COD, oils, fats and sulfates) was 78.7–89.7% after 96 h of microbial treatment.

In another study, bioremediation of Greywater contaminated with pesticides (carbendazim and thiamethoxam) and COD, was carried out under the action of *Aspergillus versicolor* fungal cells, a bacterial consortium obtained from kitchen waste sludge, and a bacterial consortium with added fungi [106]. The efficiency of degradation of COD and pesticides by the combined consortium was 93.6–94.4% for 10 days, whereas the same parameter was 44–54% and 84.8–87.9%, respectively, for the bacterial consortium and the individually applied fungal cells. Thus, the advantages of combining bacterial and fungi were obvious.

To clean real textile effluents, a fungal consortium consisting of two microorganisms (*Aspergillus flavus* and *Fusarium oxysporium*) was used. After 7 days, the effluent degradation efficiency was 78%, while the same parameter for individual crops was 52.4–54.7%. The effectiveness of wastewater treatment was not only assessed by removing dyes, but also by reducing COD, which amounted to 77.6% [107].

In another study, the effectiveness of degradation of textile effluents was evaluated by a bacterial yeast consortium (*Brevibacillus laterosporus* and *Galactomyces geotrichum*), whose cells were immobilized in a Ca-alginate gel or a mixed gel of alginate and poly(vinyl alcohol) [108].

Immobilized cells were able to degrade wastewater for five cycles with an efficiency of 74–95%. Cells of the same consortium, immobilized on a stainless-steel carrier, provided 90% of wastewater degradation in a reactor with continuous operation at a flow rate of 10 mL/h for three operating cycles [109].

A mixed consortium, including cells of the bacteria *Ralstonia pickettii* and the fungi *Trichoderma viride*, was used in the biodegradation of chlorobenzene. At an initial concentration of 220 mg/L, this compound was completely removed from the medium for 60 h. The degree of chlorobenzene mineralization, estimated by CO_2_ yield, was 86.3%. The same 100% removal of the pollutant was much slower when the fungi was used alone (96 h) or the bacterial cells (72 h). It is interesting to note that this consortium was able to utilize 1000 mg/L chlorobenzene completely during 105 h. Intermediates of chlorobenzene degradation, which inhibit bacterial growth, were easily metabolized by fungal cells, allowing bacteria to continuously participate in the joint biodegradation process [106].

The bioprocessing of poly(vinyl acetate) containing wastewater was carried out under the action of cells of a fungal consortium (*Chaetomium globosum*, *Aspergillus niger* and *Rhizopus oryzae*) [110]. The degree of COD removal and degradation was 97.8% and 99.8%, respectively. Enzymatic analysis showed that extracellular laccase and lignin peroxidase participated in biodegradation. In addition, the analysis of metabolic products showed their mineralization with the formation of formic acid and ethanol.

Simultaneous removal of phenol (1200 mg/L) and selenite (10 mg/L) from synthetic wastewater was investigated using a consortium composed of *Phanerochaete chrysosporium* fungus cells and *Delftia lacustris* bacteria [111]. It was found that under the action of such a consortium, selenite ions are biologically reduced to Se(0) nanoparticles with simultaneous degradation of phenol. In addition, it was found that, over time, bacteria grow and form a biofilm on the surface of the fungal cells, which does not lead to an improvement, but rather to a decrease in the efficiency of phenol decomposition. The consortium was already completely inhibited at a phenol concentration of 200 mg/L.

Thus, the bacterial biofilm limited the mass transfer and respiratory activity of the fungal culture, generally worsening the functioning of the consortium over time. Therefore, it is necessary to note here the revealed fact that the characteristics of the consortium depend on the duration of its operation.

It is obvious that directed association of microorganisms based on the preliminary selection of active pollutant destructors is an effective approach to the creation of artificial consortia. When compared to pure cultures, consortia have shown better biodegradation of specific pollutants or their mixtures that have a complex composition. The presence of fungal strains in consortia makes it possible to degrade a wider range of pollutants, which is due to the secretion of enzymes reacting with pollutions. In addition, it is possible to increase the efficiency of the process by using consortia in an immobilized form.

Immobilized consortia are attractive for use in multiple work cycles, and consortia of fungi with microalgae have a high potential in the use of pharmaceutical agents in wastewater treatment [93]. This approach to solving the problems of pollutant removal and the bioremediation of media is, in comparison to methods of joint cultivation of microalgae and bacteria, still relatively insufficiently studied.

## 3. Analysis of Current Trends in the Development of Fungal-Containing Consortia

### 3.1. Genetically Modified Microorganisms in Artificial Consortia with Fungi

Despite the high efficiency of degradation of various pollutants by natural microbial consortia [112], there has recently been a growing demand for strains that are improved by using advanced methods of synthetic biology and metabolic engineering, and numerical methods in the field of genetic engineering [113]. To date, the use of genetically modified strains in mixed microbial consortia for the decomposition of various pollutants is recognized (Figure 1). The percentage of such consortia is comparable to variants composed of yeast and bacterial cells (Figure 1). One example of the use of such an artificial consortium, as described in Table 1, Table 2, Table 3, Table 4, Table 5, Table 6 and Table 7, is the heterologous expression of genes encoding MnP and LiP enzymes in non-ligninolytic fungi, which made it possible to complement the pathway of degradation of PAHs [88], without leaving toxic intermediates in the treated medium.

However, the traditional method of random mutagenesis is a time-consuming approach, and searching for key amino acid mutations is extremely complex and requires operating across a large space. With the help of computer modeling, it is possible to “calculate” enzymes with improved binding affinity and greater specificity of action in relation to the substrates destruction, which are both necessary [114]; it is also necessary to successfully introduce producers of “improved” enzymes into consortia, which is another serious task that confronts researchers.

Another difficulty is that genetically modified strains often have to survive high concentrations of pollutants [113,115]. Several strategies have been used to improve cellular tolerance, including changing the composition of membrane lipids, phenotypic screening through adaptive laboratory evolution, modification of global gene expression, genome shuffling, directional evolution, and others.

Although genetically modified strains are effective in bioremediation processes [11], their use is limited to laboratory studies due to minimizing the risks of environmental impact. Since there are no globally accepted regulatory documents that concern the spread of genetically modified organisms, the regulation of the development and release of genetically modified organisms varies in different countries, depending on the purposes of their use, extending from a complete ban on their import, release or use to allowing their use, subject to varying degrees of regulation. However, despite this, methods of genetically engineering filamentous fungi continue to be actively developed and used in research around the world, making it possible to overcome many of the shortcomings of classical methods for improving strains [115,116,117]. To eliminate the negative effect of such cells, it is possible to use several genetic tools, including cell self-destruction systems [118]. Such approaches to realization of programmed cell death in a certain period of their functioning can be used and activated after the completion of bioremediation or after accumulation of certain concentrations of the cells.

### 3.2. Role of Composition in Artificial Consortia with Fungal Cells

Microbial consortia have a high self-organization, which allows them to carry out the catalytic conversion of substrates with high efficiency, providing an intensive exchange of metabolites. Moreover, in consortia, cells have higher adaptability and viability in relation to environmental factors (pH, temperature, concentration of pollutants, etc.) [119]. Despite this, most artificial microbial consortia face problems of non-sustainable functioning [120]. Even minor fluctuations in the composition and activities of consortia can have a negative impact on the effectiveness of the processes taking place with their participation. The difficulty of managing the bioremediation characteristics of artificial consortia is actually determined by the qualitative (Figure 1) and quantitative composition (Figure 2) of the participants in the artificial biosystems being formed.

As the analysis of the data in Table 1, Table 2, Table 3, Table 4, Table 5, Table 6 and Table 7 shows that, despite the diversity of the consortia being developed (Figure 1), the ones most widely used to create effective bioremediation microbial systems, that have demonstrated their advantages over natural systems, consist of two or three microorganisms, of which one is a fungal culture (Figure 2). The maximum part of all artificial consortia currently being developed combines mycelial fungi with each other and with cells of different bacteria (Figure 1).

It is known that fungi and bacteria are characterized by different rates of synthesis of enzymes that are necessary for the catalysis of different processes in the bioremediation of pollutants. Consequently, the rates of different processes may be compared by varying the biomass of certain cells introduced into heterogeneous consortia in order to ensure their maximum effectiveness of action. This is the basis for the unification of certain microorganisms into artificial consortia (Figure 3).

After analyzing the data in Table 1, Table 2, Table 3, Table 4, Table 5, Table 6 and Table 7, it is possible to conclude that there are consortia consisting of two–three strains of various fungi, mainly mycelial types, that are most effective in removing 70–90% of heavy metals in mixtures from various media.

For the removal of dyes, the most effective are the consortia consisting of yeast cells, which provide bioremediation efficiency of up to 100%; the destruction of pesticides is the most successful under the action of consortia consisting of fungal cultures, such as *Verticilium* sp. and *Metacordyceps* sp.

Synthetic polymers undergo the most difficult microbial degradation, but the maximum weight loss of synthetic polymers (microplastics) is achieved under the action of consortia that combine bacteria, mycelial fungi or yeast.

In the case of the presence of mixtures of several PAHs in media subjected to bioremediation, the most effective (in terms of the degree of degradation of the pollutant (86–100%)) were consortia that included bacteria and white rot fungus *Pleurotus ostreatus*, as well as an immobilized consortium consisting of the bacteria *Kocuria rosea* and the fungi *Aspergillus sydowii*.

Among the most effective in the removal of pharmaceutical pollutants were selected consortia from cells of white rot saprobic fungus *Pycnoporus sanguineus* and *Phanerochaete chrysosporium* (the removal efficiency of ciprofloxacin-, norfloxacin and sulfamethoxazole in their mixture was 100%), as well as tinder fungi *Ganoderma applanatum* and *Laetiporus sulphureus* (efficiency of degradation of a mixture containing celecoxib, diclofenac and ibuprofen was 99.5%).

Bacterial and fungal consortia proved to be the most effective for the treatment of real wastewater from industrial enterprises and oil pollution. The fungal consortium consisting of *Phanerochaete chrysosporium* and *Delftia lacustris* can be used to remove phenol, with an efficiency of over 90%.

Of course, in order to achieve the effective functioning of synthetic microbial consortia, it is important to predict the possible types of interactions of all microorganisms involved in the functioning of artificially created biosystems [35,36].

Most often, these interactions are based on mutualism or competition [13,121]. Mutualism suggests that jointly cultivated microorganisms have a beneficial effect on each other, while the composition of artificial consortia is stabilized, but only at a certain cell density [34,111].

In case of competition for a substrate, consortium members can release metabolites into the environment that negatively affect other consortium members (organic acids, mycotoxins, antibiotics, antimicrobial peptides, enzymes) [122,123,124,125,126]. As a rule, these processes are regulated by cell Quorum Sensing (QS) [16]. QS molecules and the conditions of their formation can be used to control or regulate the expression of certain genes, control the composition of consortia, and ensure intercellular connections between certain consortium members [34,120,127]. In considering this, modern studies of artificial consortia should enable the creation of model systems for the accumulation of toxic metabolites of a number of microorganisms (fungi, microalgae, bacteria) and develop effective ways to detoxify them.

Cell immobilization can also solve the instability problems of microbial consortia, since cellular communication can be more active in a confined space and positively affects the speed of bioremediation processes. At the same time, the inclusion of cells in gel matrices [52,77,108] simulates the development of a stabilized state of cells and the additional formation of biofilms [11,34,91,103,111].

Immobilized fungal consortia can be used in non-sterile conditions at a separate stage, and then be integrated into conventional waste treatment systems before the action of aerobic sludge at water processing stations [128,129,130]. In some cases, the self-stabilization of the artificial fungi-containing consortia is sufficient for their use in non-immobilized form in laboratory conditions to treat various real contaminated industrial and environmental water and soil samples [39,44,49,51,94,107,108,110]. While such studies bring the introduction of fungal consortia closer to practice, there are not many known examples of pilot or industrial tests of artificial consortia [131,132,133]. Some of them follow:-microbial consortium containing *T. versicolor*, *P. ostreatus*, *Phanerochaete* sp., *Pseudomonas fluorescens* and *B. subtilis* cells was applied for the treatment of non-domestic wastewater. This fungal/bacterial consortium was prepared by mixing fungal biomass pellets with suspensions of bacterial cells. The removal of colored substances (2700 Color Units_550nm_), COD (1.75 g/L) and nitrate (3 mg/L) was 91 ± 2%, 90 ± 4% and 17 ± 2%, respectively, after 15 days of water treatment at a pilot plant [131];-consortium of *A. niger*, *Mucor hiemalis* and *Galactomyces geotrichum*, has been tested for the treatment of real wastewater from industry at a pilot scale station (110 L) and industrial wastewater treatment plant (1000 L). The efficiency of COD removal in the industrial reactor was 50% under the influence of this consortium [132];-consortium containing *Acinetobacter oleivorans*, *Corynebacterium* sp., *Pseudomonas* sp, *Rhodococcus* sp., *Micrococcus* sp. and yeast *Yarrowia* sp. was tested by Ecophile Co., Ltd. (Korea) in the biodegradation of hydrocarbons in soil (2300 mg/kg) contaminated with diesel fuel. This large-scale experiment involved two samples of 100 metric tons of contaminated soil, both without (control) and with consortium treatment (10^9^ cells/kg of soil). The introduction of consortium reduced pollution by 57.7% within 2 weeks, whereas in the control (without the consortium), degradation was only 10.1% [133].

Thus, such positive samples of scaling use of artificial fungal consortia not only demonstrate their real-world efficacy, but also addresses potential solutions encountered during practical applications.

## 4. Conclusions

Today the question of the action efficiency of artificial fungi-containing consortia in the bioremediation of various pollutants in real conditions remains open and continues to be studied, despite the accumulating information in proven positive laboratory studies of their functioning. The stabilization of the complex composition and functional characteristics of created multicomponent biosystems is one of the key points for the successful scientific and practical solution of problems in the field of creating new artificial consortia, including those combining fungi with similar and other types of microorganisms. The analysis of the accumulated results, their generalization and the identification of common positive approaches that combine certain microorganisms and organize their use can form the basis for the development of new processes for the removal of various pollutants that, with the highest possible level of efficiency, bioremediate various media. The economic viability and attractiveness of using artificial consortia containing fungi for bioremediation determines the cost-effectiveness of cultivating individual strains and their availability for commercial application. The availability can be satisfied by searching for and isolating the necessary fungi from natural sources or by creating genetically modified strains [116,117]. Whereas the costs of obtaining the necessary amounts of biomass of fungal cells are, according to the economic estimates just conducted, considered low enough [116,134]. Enzymes synthesized by fungi, the main catalysts of many redox and hydrolytic processes, are already commercially available, although they are more expensive products, when compared to the fungal cells producing them. This should be noted when estimating the economic viability and scalability of using artificial fungi-containing consortia in large-scale bioremediation projects. The positive aspects of the economy associated with the use of fungi in various processes, including bioremediation [135], may contribute to the wider use of artificial consortia with fungi in the near future.

## Figures and Tables

**Figure 1 microorganisms-12-00470-f001:**
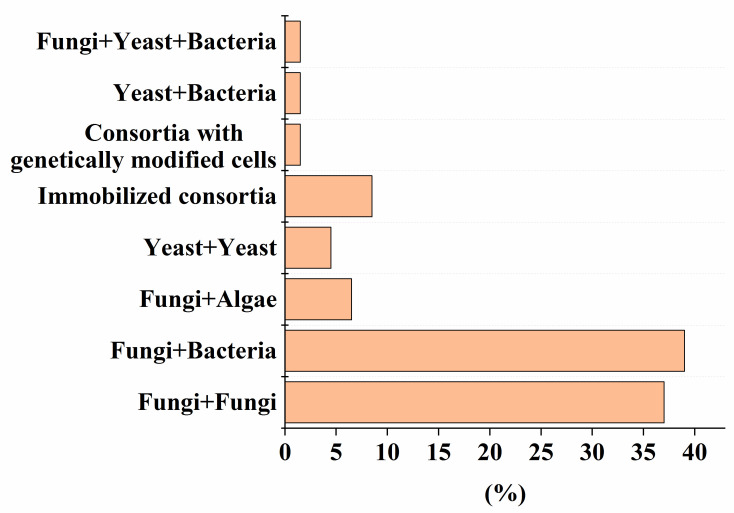
The percentage of artificial consortia of different composition of the total number (56) of reviewed studies, presented in Table 1, Table 2, Table 3, Table 4, Table 5, Table 6 and Table 7.

**Figure 2 microorganisms-12-00470-f002:**
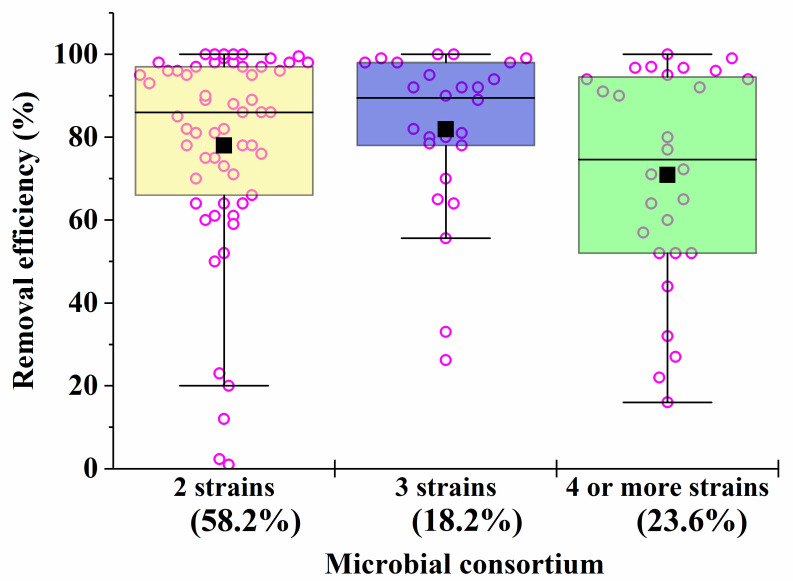
The average efficiency of degradation of pollutants by consortia with two, three, four and more strains in the composition. The percentage of the total number of all options presented in Table 1, Table 2, Table 3, Table 4, Table 5, Table 6 and Table 7 (56 studies reviewed) was calculated. Each point corresponds to the research results listed in Table 1, Table 2, Table 3, Table 4, Table 5, Table 6 and Table 7.

**Figure 3 microorganisms-12-00470-f003:**
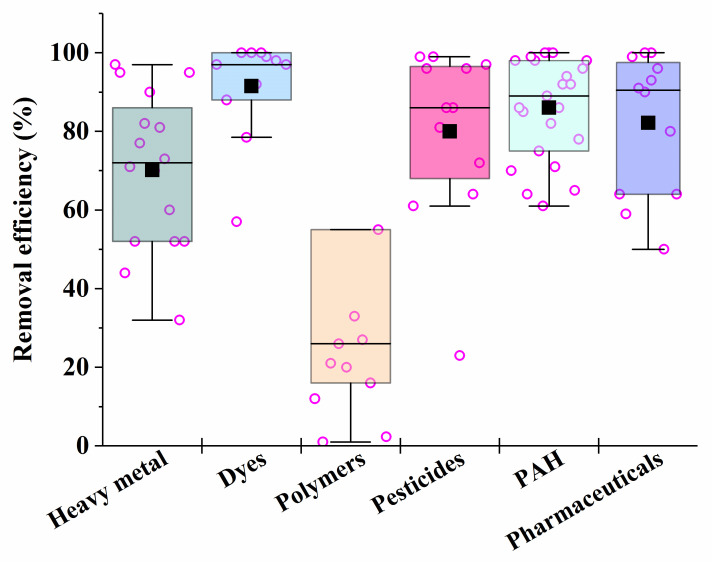
The average efficiency of the consortia for elimination of different types of pollutants, calculated on the basis of sources performed in Table 1, Table 2, Table 3, Table 4, Table 5, Table 6 and Table 7. Each point corresponds to the research results listed in Table 1, Table 2, Table 3, Table 4, Table 5, Table 6 and Table 7. The six differently colored squares correspond to the six pollutants shown on the abscissa axis.

**Table 1 microorganisms-12-00470-t001:** Microbial consortia with fungal species for removal of heavy metals.

Consortia [Reference]	Conditions	Pollutant/Process Efficiency
*Aspergillus niveus*, *A. flavus*, *A. niger* [39]	1.4 × 10^6^ spore/mL of each strain; pH 5.0, 110 rpm, 30 °C, 96 h	Removal of Cr, Zn, Pb, Cd and Ni—70–90%
*A. flavus*, *A. fumigatus* [40]	Heavy metal concentration—100 mg/L, 1.2 × 10^6^ spores/mL, pH 5.0, 30 °C, 144 h	Removal of Cr(VI)—81%, Cd(II)—82%, mixture of metals—73%
*Ascomycota* and *Basidiomycota* fungi [41]	Initial metal concentration (23–2347 mg/kg), pH 7.9, soil moisture 60–65%, 28 °C, 100 days	Removal of As—77%, Cr—60%, Cu—52%, Fe—52%, Mn—71%
*Ascomycota* and *Basidiomycota* fungi [42]	Initial metal concentration (400–800 mg/kg), pH 7.9, soil moisture 60–65%, 28 °C, 100 days	Removal efficiencies of Ni, Pb, Zn—52%, 44%, 32% respectively
*A. fumigatus*, *A. terreus*, *Paenibacillus dendritiformis* [43]	Cd—100 mg/L, pH 5.0, 30 °C, 120 h	Removal of Cd(II)—95%
*A. terreus*, *Talaromyces islandicus*, *Neurospora crassa*, *Aspergillus flavus* [44]	Pb(II)—20.5–293.23 mg/L, Ni(II)—12.1–164.7 mg/L, inoculum 8%, pH 5.0, 30 °C 120 h	Removal of Pb(II) and Ni(II)—95–97%

**Table 2 microorganisms-12-00470-t002:** Microbial consortia with fungal species for decolorization of dyes.

Consortia [Reference]	Conditions	Pollutant/Process Efficiency
*Yarrowia* sp., *Barnettozyma californica*, *Sterigmatomyces halophilus* [49]	100 mg/L of dye,30 °C, static conditions, 6–12 h	Degradation of Scarlet GR, Red HE3B, Remazol Brilliant Blue R, Methyl Orange, Rubine GFL and Reactive Red 2—92–100%
*Daldinia concentrica*, *Xylaria polymorpha* [50]	50 mg/L of dye, pH 4.5, 30 °C, 150 rpm, 48 h.	Degradation of cibacron brilliant red 3B-A—99%
*Rhodotorula* sp., *Raoultella planticola* and *Staphylococcus xylosus* cells immobilized in Ca-alginate beads [51]	200 mg/L of methylene blue in municipal wastewater and industrial effluent, 144 h	Degradation of methylene blue—100% and 78.5% in municipal wastewater and industrial effluent, respectively
*A. niger*, *A. terrus*, *A. oryzae*, *A. fumigatus* [52]	20 mg/L of each dye, 150 rpm, 28 °C, 72 h	Degradation of reactive blue 4, fast green, methyl red, crystal violet, alura red AC, tartrazine, naphthol blue black, janus green B, alizarin yellow R, evans blue, brilliant green, pararosaniline, ponceau S, cibacron brilliant red 3B-A, direct violet 51—57–100%
*Aspergillus* sp., *Chlorella sorokiniana* [53]	Disperse Red—0.1 g/L, pH 6.0, 160 rpm, 25 °C, 4 days	Degradation/adsorption of disperse red 3B—98.1%
*Daedalea dickinsii*, *Pseudomonas aeruginosa* [54]	Methyl orange—100 mg/L, 30 °C, 7 days	Degradation of methyl orange—98%
*Sterigmatomyces halophilus*, *Meyerozyma guilliermondii* [55]	Reactive Black 5, Acid Orange 7; Reactive Green 19, Reactive Yellow, ABC, Atlantic Black C—50 mg/L, glucose as co-substrate, pH 7.0, 35 °C, 120 h	Degradation—88–97%
*Penicillium oxalicum*, *Aspergillus tubingensis* [56]	100 mg/L of congo red with dextrose (10 g/L), pH 5, 150 rpm, 28 °C, 12 h	Congo red degradation—97.1%

**Table 3 microorganisms-12-00470-t003:** Microbial consortia with fungal species for degradation of synthetic polymers.

Consortia [Reference]	Conditions	Pollutant/Process Efficiency
*Sterigmatomyces halophilus*, *Meyerozyma guilliermondii*, *M. caribbica* [60]	30 °C, 45 days	Low-density polyethylene (LDPE) mass reduction—33.2%
*A. niger*, *P. aeruginosa* [61]	37 °C, 30 days	Polyurethane weight loss—20%
*Curvularia lunata*, *Alternaria alternata*, *Penicillium* *simplicissimum*, *Fusarium* sp. [62]	90 days	Polyethylene weight loss—27%
*A. niger*, *A. flavus*, *A. oryzae* [63]	55 days	Polyethylene weight loss—26.2%
Microorganisms isolated from activated sludge and river sediments (*Lysinibacillus massiliensis*, *Bacillus licheniformis*, *B. indicus*, *B. megaterium*, *B. cereus*, *Pseudomonas alcaligenes*, *Aspergillus* sp., *Penicillium* sp., *Alternaria* sp., *Candida parapsilosis* [64]	160 rpm, 56 days at room temperature, 10 mL of bacterial and fungi suspension, and one film sample (1 cm^2^) of polymer materials	Weight loss of sample (LDPE & thermoplastic starch & styrene-ethylene-styrene)—16%
Microorganisms isolated from compost (*B. sonorensis*, *B. subtilis*, *Aspergillus* sp. *Trichoderma* sp., *Rhizopus* sp.) [64]		Weight loss—21.9%
Microorganisms of enriched landfill soil (*Achromobacter xylosoxidans*, *Trichosporon chiropterorum*, *Penicillium chalabudae)* [65]	pH 7.2, 150 rpm, 30 °C, 90 days	LDPE weight loss—55.6%
*Aspergillus* sp., *Penicillium* sp. [66]	29 °C, 85% humidity, 30 days	Polypropylene/poly (butylene adipate-*co*-terephthalate)/thermoplastic starch weight loss—1.0–2.3%
*Bacillus* sp., *Aspergillus* sp. [67]	30 °C, 150 rpm, 30 days	LDPE weight loss—12%

**Table 4 microorganisms-12-00470-t004:** Microbial consortia with fungal species for degradation of pesticides.

Consortia [Reference]	Conditions	Pollutant/Process Efficiency
*Fomitopsis pinicola*, *B. subtilis* [74]	30 °C, 7 days	DDT (1,1,1-trichloro-2,2-bis(4-chlorophenyl) ethane) degradation—86%
*Pleurotus ostreatus*, *P. aeruginosa* [75]	25 °C, 7 days	DDT degradation—86%
*A. niger*, *Chlorella vulgaris* [76]	38 pesticides in mixture—total concentration—72.7µg/L, biomass—181.6 mg dry weight/L, pH 4.0, 100 rpm, 68 h	Degradation—23%
*Verticilium* sp., *Metacordyceps* sp. [77]	Concentration of each pesticide—50 mg/L, 100 rpm, pH 5.5, 27 °C, 21 days	Degradation of atrazine—81%, iprodione—96%; chlorpyrifos—99%
*Verticilium* sp., *Metacordyceps* sp. immobilized in Ca-alginate beads [77]	Concentration of each pesticide—50 mg/L, flow rate—90 mL/h, inoculum concentration—30 *w/v*, 100 rpm, 27 °C	Degradation of atrazine—64%, iprodione—96%; chlorpyrifos—85% (11–15 days)
Consortium of microorganisms present in coconut fiber, garden compost and agricultural soil and *Trametes versicolor* [78]	Mixture of pesticides—30–40 mg/kg, pH 6.4, 25 °C, 16 days	Degradation of atrazine—72.2%, carbendazim—96.7%, carbofuran—98.7%, metalaxyl—96.7%
*Fomitopsis pinicola*, *Ralstonia pickettii* [79]	DDT—5 mM, 30 °C, 7 days	DDT degradation—61%

**Table 5 microorganisms-12-00470-t005:** Microbial consortia with fungal species for degradation of PAHs.

Consortia [Reference]	Conditions	Pollutant/Process Efficiency
*Acremonium* sp, *B. subtilis* [85]	Concentration of each PAH in mixture—50 mg/L, 28 °C, 160 rpm, 10 days	Degradation of naphthalene—100%, fluorine—89%, phenanthrene—82%, anthracene—71%, fluoranthene—61%
*Pleurotus ostreatus*, *Penicillium chrysogenum* [86]	30 °C, 30 days	Degradation of benzo[a]pyrene—86%
*P. ostreatus*, *P. aeruginosa* [86]		Degradation of benzo[a]pyrene—75%
Consortium (*Proteobacteria*, *Bacteroidota*, *Fusarium*) immobilized on biochar [87]	Mixture of 50 mg/L of phenanthrene and 150 mg/L of Cd^2+^, 150 rpm, 30 °C, 7 days	Degradation of phenanthrene—92–98%, removing of Cd^2+^—94–99%
Consortium with two genetically modified strains of *A. niger* [88]	Mixture of pyrene and benzo(a)pyrene—1000 mg/kg soil, pH of 8.4, 30 °C, 14 days	Degradation efficiency of phenanthrene—92%, pyrene—64%, benzo(a)pyrene—65%
*Kocuria rosea* and *A. sydowii* immobilized in guargum-nanobentonite composite water dispersible granules [89]	Mixture of naphthalene, fluorene, phenanthrene, anthracene, and pyrene—100 µg of each PHA/g of soil, pH 8.3, 27 °C, 30 days	Degradation efficiency—85–100%
*P. putida*, yeast *Basidioascus persicus* [90]	800 mg/L of pyrene, rhamnolipid biosurfactant 100 μL, 28 °C, 21 days	Degradation efficiency—78%
*Ochrobactrum intermedium* and white rot fungus *Pleurotus ostreatus* [91]	Concentrations of different PAH—138.2–268.0 mg/kg of soil, moisture—70%, 30 °C, 110 days	Degradation of fluoranthene, indene[1,2,3-cd]pyrene and benzo[g,h,i]perylene—100%; Anthracene, pyrene, chrysene and benzo[a]anthracene—96%, 86%, 98% and 98%, respectively
*Pleurotus ostreatus*, *Azospirillum brasilense* [92]	A mixture of anthracene, phenanthrene, fluorene, pyrene, and fluoranthene—50 mg/L, 130 rpm, 24 °C, 14 days	Degradation efficiency > 70%

**Table 6 microorganisms-12-00470-t006:** Microbial consortia with fungal species for degradation of pharmaceutical pollutants.

Consortia [Reference]	Conditions	Pollutant/Process Efficiency
*Pycnoporus sanguineus*, *Phanerochaete chrysosporium* [95]	Each antibiotic concentration—10 mg/L, biomass of each strain—0.15 g dry weight/L), pH 4.5, 30 °C, 4 days	Removal efficiency of ciprofloxacin, norfloxacin and sulfamethoxazole in their mixture—100%
*Pycnoporus sanguineus*, *Alcaligenes faecalis* [96]	Sulfamethoxazole (50 mg/L) and vitamins mixture (VB2, VB6, VB12 and VC), 28 °C, 120 rpm, 24 h	Sulfamethoxazole degradation—93%
*Ganoderma applanatum*, *Laetiporus sulphureus* [97]	Concentration of each of three pollutants—10 mg/L, pH 6.4, ambient temperature, 150 rpm, 72 h	Degradation (mixture of celecoxib, diclofenac and ibuprofen)—99.5%
*A. niger*, *Mucor circinelloides*, *Trichoderma longibrachiatum*, *Trametes polyzona* and *Rhizopus microsporus* [98]	Pollutants concentration—1 mg/L, pH 4.6, 30 °C, 7 days, consortium concentration—30% (*v*/*v*)	Degradation of carbamazepine—90%, diclofenac sodium—96% and ibuprofen—91%
*A. niger*, *C. vulgaris* [99]	Pharmaceutical substances—8–11 μg/L, microalgae-fungus biomass—75 mg dry weight/L, 72 h	Relative removal of initial ranitidine concentrations—50%
*Penicillium rastrickii*, *P. oxalicum*, *Cladosporium cladosporoides*, *Micrococcus yunnanensis*, *Oligella ureolytica*, *Sphingobacterium jejuense* [100]	Mixture of diclofenac, carbamazepine and ketoprofen with 100 μM of each compound, 28 °C, 10 days	Degradation of diclofenac—99%, ketoprofen—80%
*Chlorella vulgaris*, *Aspergillus oryzae* [101]	Simulated swine wastewater with addition of 0.1–0.5 mg/L Cu (II), 0.4 mg/L of mixture of antimicrobial agents, pH 7.2, 28 °C, 14 days	Removal efficiency of sulfamonomethoxine, sulfamethoxazole and sulfamethazine—58.8%, 63.5%, and 63.9%, respectively

**Table 7 microorganisms-12-00470-t007:** Microbial consortia with fungal species for degradation of various pollutants not mentioned in Table 1, Table 2, Table 3, Table 4, Table 5 and Table 6.

Consortia [Reference]	* Conditions	Pollutant/Process Efficiency
*Acinetobacter baumannii*, *Talaromyces* sp. [102]	The initial concentration of petroleum in soil—1220 mg/kg, pH 8.3, 30 °C, 28 days	Degradation of petroleum—65.6%
*Paraburkholderia* sp., *Paraburkholderia tropica*, *Scedosporium boydii* [103]	1% *v/v* crude oil, 120 rpm, 30 °C, 7 days	Degradation of crude oil—81.5%
*Scedosporium* sp., *Acinetobacter* sp. [104]	Crude oil—200 mg/L, pH 7.0, 150 rpm, 30 °C, 7 days	Crude oil degradation—58.6%
*Micrococcus luteus*, *Rhodococcus equi*, *A. niger* [105]	Greywater—COD—1165.6 mg/L, oil and grease—58 mg/L, sulphate—95.6 mg/L, pH 7, 35 °C, 96 h	Degradation of COD, oil and grease and sulphate were 78.7, 82.6 and 89.7%, respectively
*Aspergillus versicolor* and bacterial species (*Pseudomonas*, *Klebsiella* species, *B. subtilis*) [106]	Greywater with 100 μg/L of carbendazim and thiamethoxam, 80 rpm, 30 °C, 240 h	Degradation of carbendazim and thiamethoxam 94.4 and 93.6%, respectively
*A. flavus*, *Fusarium oxysporium* [107]	Real textile effluent pH 8.7, COD—611 mg/L, pH 6.0–8.0, 28 °C, 7 days	Degradation—78.1%, COD removal—77.6%
Consortium of *Brevibacillus laterosporus* and *Galactomyces geotrichum* immobilized into Ca-alginate or polyvinyl alcohol-alginate beads [108]	Textile industry effluent pH 8.8, COD—2400 mg/L, 48–60 h	Degradation during 5 repeated cycles—76–95%
*Ralstonia pickettii*, *Trichoderma viride* [109]	Chlorobenzene—220 mg/L, 160 rpm, 28 °C, 60 h	Chlorobenzene degradation—100%
*Chaetomium globosum*, *A. niger*, *Rhizopus oryzae* [110]	Poly(vinyl acetate) processing wastewater pH 7.1, COD—23.48 g/L; pH 5.5, 150 rpm, 28 °C, 10 days	COD, poly(vinyl acetate) and color removal yields—97.8%, 98.5% and 99.8%, respectively.
*Phanerochaete chrysosporium*, *Delftia lacustris* [111]	Phenol (1000 mg/L) and selenite concentration—10 mg/L, 180 rpm, pH 6.5, 30 °C, 120 h	Phenol degradation—97.8% with the simultaneous reduction of selenite to Se(0)

* COD—chemical oxygen demand.

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
