# Peer review of "Using Fungi in Artificial Microbial Consortia to Solve Bioremediation Problems"

_microorganisms, 2024, doi:10.3390/microorganisms12030470_

Round 1
Reviewer 1 Report
Comments and Suggestions for Authors
The study's objective is intriguing, emphasizing the consortium as the protagonist rather than the pollutant. However, there is a lack of foundational discussion on consortia, especially in the final part of the article. In the section where the authors discuss composition, I believe many points should be raised, including "One of the challenges in standardizing a fungal consortium is the characteristic of mycelial growth, especially in non-sporulating fungi. In this sense, it is important for this work to provide a brief discussion on this aspect." As well as discussing the forms of standardization and control of the consortia, which are briefly mentioned at one point.
Therefore, I suggest improving the final part; it seems like a very immature discussion of the potential that the article holds.
Line 33. Biodestructors" is not a commonly used term in the scientific literature within the field of microbial biotechnology. The more prevalent terms for describing microorganisms or processes that break down or degrade organic substances are "biodegraders" or "biodegrading microorganisms.
Line 38. Please rephrase the sentence: “of pro- and eukaryotic microorganisms (in particular, filamentous fungi and yeast)”, as the original placement of prokaryotes felt awkward, taking the example of the association between filamentous fungi and yeast."
Line 82. Rephrase the sentence: …“chemical similarity of signaling (for example, lactone-containing) quorum molecules in cells of different microorganisms" …
Line 95. Including a brief introduction based on research from international patent databases, there is some information regarding patented consortia.
Line 105. I suggest an introduction on how the heavy metal treatment process occurs, addressing the issues of adsorption and accumulation.
Line 150. The sentence raises a few points. 1. It mentions mushrooms, but the microorganisms in the article are filamentous fungi and bacteria. 2. The text needs to clarify who the natives are and identify the exogenous microorganism
Line 160. Reformulate the partial conclusion, particularly explaining the issue of unequivocal use and providing a clearer explanation of foreign fungi. Additionally, I have doubts about the use of the term “foreign”; typically, we use native and exogenous fungi.
Line 174. Please include the information that dyes are one of the most studied pollutants for degradation by fungi, with hundreds of articles published in the field.
Line 176. The degradation of dyes differs from discoloration. Discoloration, for instance, can occur due to accumulation and adsorption, not necessarily because of degradation. Please review these terms and improve the text.
Line 181. I didn't understand why 'yeast consortium' was included in the paragraph. It seems out of place. It might be necessary to clarify its connection with the rest of the text or consider revising it to maintain paragraph cohesion.
Line 185. The text in this entire section became confusing. I suggest distinguishing between yeasts, sporulating fungi, and basidiomycetes. I missed a discussion about the importance of basidiomycetes in this process. They are the group with the highest enzymatic activity and are widely used in dye treatment. However, a question remains: Are there consortia involving basidiomycetes? Please review and elaborate on this point.
Line 256. Conduct a brief discussion on which polymers were considered in your review, the names, and the characteristics. Briefly explain the difference in polymer sizes and what is considered microplastics.
Line. 329. Explain further the advantage of using consortium for pesticide degradation, compared to a single isolate. Explain the issue of metabolites produced in the process and the syntrophic action of microorganisms.
Line 378. Many associations are between bacteria and fungi, explain why.
Line 477. Reference
Line. 532. Explain further the issue of what you considered as 'negatively acting species'.
Line 551. The paragraph seems to be lost; it's not clear whether it relates to the study of the previous paragraph.
Line 653. The paragraph is confusing; the issue of HPAs is unclear. I don't know if the survey presented in the graph is only for this pollutant, for example. Rewrite.
Line 661. Figure 1. Add more information to the caption about the data that comprised the analysis. For example, indicate the number of trials.
Line 695. Figure 2. As with the previous figure, it lacked detail on which articles were selected and why they were chosen. A suggestion for the figures would be to add, as supplementary material, the list of articles used.
Line 711. Place after the figure.
Line 722. From line 722 onwards, there is repeated information as it has already been presented in Figure 3, and the reader can quickly grasp this. I suggest transforming some parts into conclusions and improving the conclusion, which was too superficial.
Comments on the Quality of English Language
The quality of the English is not bad, however, some expressions uncommon in the field are used, which may cause confusion to the reader. Perhaps reviewing the translation of these points would be beneficial.
Author Response
Dear Reviewer,
We are grateful to you for the suggestions allowing us the improving of our manuscript.
Please, see our comments to your remarks and the revised text of the paper:
The study's objective is intriguing, emphasizing the consortium as the protagonist rather than the pollutant. However, there is a lack of foundational discussion on consortia, especially in the final part of the article. In the section where the authors discuss composition, I believe many points should be raised, including "One of the challenges in standardizing a fungal consortium is the characteristic of mycelial growth, especially in non-sporulating fungi. In this sense, it is important for this work to provide a brief discussion on this aspect." As well as discussing the forms of standardization and control of the consortia, which are briefly mentioned at one point. Therefore, I suggest improving the final part; it seems like a very immature discussion of the potential that the article holds.
We sincerely thank you for your comments and suggestions to improve the quality of our article. We have tried to take into account all of them and made specified corrections to the text. We have also edited the English language throughout the article to simplify all complex sentences and make it clearer as much as possible.
Line 33. Biodestructors" is not a commonly used term in the scientific literature within the field of microbial biotechnology. The more prevalent terms for describing microorganisms or processes that break down or degrade organic substances are "biodegraders" or "biodegrading microorganisms.
It has been corrected.
Line 38. Please rephrase the sentence: “of pro- and eukaryotic microorganisms (in particular, filamentous fungi and yeast)”, as the original placement of prokaryotes felt awkward, taking the example of the association between filamentous fungi and yeast."
We have corrected the sentence to make it clearer.
Line 82. Rephrase the sentence: …“chemical similarity of signaling (for example, lactone-containing) quorum molecules in cells of different microorganisms" …
It has been corrected.
Line 95. Including a brief introduction based on research from international patent databases, there is some information regarding patented consortia.
Thank you for your suggestion. We would like to provide some clarity about why patented consortia are not discussed in this review:
1) It is known that patents, essentially, are more of a legal document than a scientific one and contain more theoretical research. Thus, they allow copyright to be assigned to researchers.
2) In this regard, patents are often presented in a general way and it is not always possible to understand the exact composition of consortia leading to a positive result (participants, ratio, etc.). For example, you can take a look at the description in these patents: Microbial consortium for wastewater treatment. US20160167996A1, 2016. (https://patents.google.com/patent/US20160167996A1/en 02/12/2024); Method for immobilizing a microbial consortium for cadmium bisorption. WO2022045906A1, 2022 (https://patents.google.com/patent/WO2022045906A1/en, 02/12/2024).
3) The total amount of such patents over the past 5-10 years is significantly less compared to the discussed in this review scientific articles, which analyzed such consortia.
4) These patents do not have a significant impact on the direction of developments in the filed of fungal consortia in recent years. While the purpose of the review was precisely to concentrate attention on this area.
The combination of all the points mentioned above allows us to conclude that the main focus of all research in the field of fungal consortia is concentrated in scientific articles.
Line 105. I suggest an introduction on how the heavy metal treatment process occurs, addressing the issues of adsorption and accumulation.
The following text has been added: “Bioremediation of heavy metals by microorganisms includes the following main mechanisms: adsorption, bioaccumulation, biotransformation and bioleaching [Zhou, B.; Zhang, T.; Wang, F. Microbial-based heavy metal bioremediation: toxicity and eco-friendly approaches to heavy metal decontamination. Appl. Sci. 2023, 13, 8439. https://doi.org/10.3390/app13148439].”
Line 150. The sentence raises a few points. 1. It mentions mushrooms, but the microorganisms in the article are filamentous fungi and bacteria. 2. The text needs to clarify who the natives are and identify the exogenous microorganism.
It has been corrected.
Line 160. Reformulate the partial conclusion, particularly explaining the issue of unequivocal use and providing a clearer explanation of foreign fungi. Additionally, I have doubts about the use of the term “foreign”; typically, we use native and exogenous fungi.
We have corrected the sentence to make it clearer and replaced the word “foreign” with “exogenous”.
Line 174. Please include the information that dyes are one of the most studied pollutants for degradation by fungi, with hundreds of articles published in the field.
The following text has been added: “Decolorization and degradation of dyes by fungi are one of the most studied processes with hundreds of articles published in the field [Kumar, V.; Singh, G.; Dwivedi, S.K. Dye degradation by fungi. dye biodegradation, mechanisms and techniques: recent advances. In: Muthu, S.S., Khadir, A. Eds.; Dye Biodegradation, mechanisms and techniques. Sustainable Textiles: Production, processing, manufacturing & nhemistry. Springer, Singapore, 2022, p. 113-140. https://doi.org/10.1007/978-981-16-5932-4_5]”.
Line 176. The degradation of dyes differs from discoloration. Discoloration, for instance, can occur due to accumulation and adsorption, not necessarily because of degradation. Please review these terms and improve the text.
The decolorization of dyes from the examples given in the Table 2 occurs mainly due to biodegradation under the action of enzymes. The exception is the consortium with microalgae Chlorella sorokiniana [Tang, W.; Xu, X.; Ye, B.C.; Cao, P.; Ali, A. Decolorization and degradation analysis of Disperse Red 3B by a consortium of the fungus Aspergillus sp. XJ-2 and the microalgae Chlorella sorokiniana XJK. RSC Adv. 2019, 9, 14558-14566. https://doi.org/10.1039/c9ra01169b], where, in addition to biodegradation, there is also a place for adsorption.
We have made appropriate changes to these terms throughout the text.
Line 181. I didn't understand why 'yeast consortium' was included in the paragraph. It seems out of place. It might be necessary to clarify its connection with the rest of the text or consider revising it to maintain paragraph cohesion.
Yeasts are eukaryotic, single-celled microorganisms classified as members of the fungus kingdom. Due to its high degradation activity towards various azo dyes, yeast has practical applications in wastewater treatment.
We have additionnaly clarified this in the text.
Line 185. The text in this entire section became confusing. I suggest distinguishing between yeasts, sporulating fungi, and basidiomycetes. I missed a discussion about the importance of basidiomycetes in this process. They are the group with the highest enzymatic activity and are widely used in dye treatment. However, a question remains: Are there consortia involving basidiomycetes? Please review and elaborate on this point.
The text discusses those consortia that are described in the Table 2. And since there are no basidiomycetes in the consortium, it means that there were no articles within the search scope that were specified.
Line 256. Conduct a brief discussion on which polymers were considered in your review, the names, and the characteristics. Briefly explain the difference in polymer sizes and what is considered microplastics.
The following text has been added: “Synthetic polymers, also known as man-made polymers, are polymers that are artificially produced by humans. Synthetic polymers can be classified into four distinct categories, namely thermoplastics, thermosets, elastomers, and synthetic fibers. Synthetic polymers are derived from petroleum oil. The most common types of synthetic polymers are: polyethylene, polypropylene, polyvinyl chloride, polystyrene, polyurethane, polyvinyl chloride, polyethylene terephthalate. Microplastics are tiny plastic particles with size less than 5 mm [ссылка 7 Efremenko, E.N.; Lyagin, I.V.; Maslova, O.V.; Senko, O.V.; Stepanov, N.A.; Aslanli, A.G.G. Catalytic degradation of microplastics. Russ. Chem. Rev. 2023, 92, 1-48. https://doi.org/10.57634/RCR5069].”
Line. 329. Explain further the advantage of using consortium for pesticide degradation, compared to a single isolate. Explain the issue of metabolites produced in the process and the syntrophic action of microorganisms.
The presence of fungi in a mixed consortium allows more efficient degradation of pollutants. This occurs due to co-catabolism, since fungal metabolic products can be used by bacterial cells.
Line 378. Many associations are between bacteria and fungi, explain why.
Thank you for your comment. Actually the explanation already provided in the text of the review (See section 3.2. Role of composition in artificial consortia with fungal cells):
“Most often, these interactions are based on mutualism or competition. Mutualism suggests that jointly cultivated microorganisms have a beneficial effect on each other, while the composition of artificial consortia is stabilized, but only at a certain cell density.
In case of competition for a substrate, consortium members can release metabolites into the environment that negatively affect other consortium members. As a rule, these processes are regulated by cell Quorum Sensing. QS molecules and the conditions of its formation can be used to control or regulate the expression of certain genes, control the composition of consortia, and ensure intercellular connections between certain consortium members”.
Line 477. Reference
It has been corrected.
Line. 532. Explain further the issue of what you considered as 'negatively acting species'.
As described in the mentioned paragraph, 'negatively acting species' refers to microorganisms that when introduced into a consortium consisting of microorganisms in mutually beneficial interactions, can lead to the interruption of these interactions or ii) . Such a “negative” influence of the introduced microorganism, in turn, leads to the ability to regulate bioremediation activity of consortia, e.g. to slow down and reduce the speed of functioning of such artificial consortia.
As described in the paragraph, 'negatively acting species' refers to the species of microorganisms, which, when introduced into a consortium consisting of microorganisms in mutually beneficial interactions, can lead to their cessation. Such a 'negative' action of the introduced microorganism, in turn, leads to the ability to regulate bioremediation activity of consortia, e.g. to slow down the speed of functioning of such artificial consortia.
We have additionally clarified it in the text.
Line 551. The paragraph seems to be lost; it's not clear whether it relates to the study of the previous paragraph.
We have rephrased the paragraph to make it clearer.
Line 653. The paragraph is confusing; the issue of HPAs is unclear. I don't know if the survey presented in the graph is only for this pollutant, for example. Rewrite.
It has been corrected.
Line 661. Figure 1. Add more information to the caption about the data that comprised the analysis. For example, indicate the number of trials.
It has been corrected.
Line 695. Figure 2. As with the previous figure, it lacked detail on which articles were selected and why they were chosen. A suggestion for the figures would be to add, as supplementary material, the list of articles used.
Thank you for your suggestion. The total number of selected articles has been clarified. All articles selected for these figures already have been cited in tables 1-7, where the specific results achieved in these studies were also indicated.
Line 711. Place after the figure.
It has been corrected.
Line 722. From line 722 onwards, there is repeated information as it has already been presented in Figure 3, and the reader can quickly grasp this. I suggest transforming some parts into conclusions and improving the conclusion, which was too superficial.
We are very grateful for your recommendations, as they allowed us to improve the quality of our review. The conclusion was expanded, the end of the generalizing part was also expanded with practically oriented information and additional attention paid to the examples of the use of artificial consortia with fungal cells. We put this new text before the conclusion. Several new references corresponding to this text were added to the list of references. The inserted text:
In some cases, the self-stabilization of the artificial fungi-containing consortia is sufficient for their use in non-immobilized form under laboratory conditions for the treatment of various real contaminated industrial and environmental water and soil samples [39,44,48,50,93,106,107,109]. Such studies bring the introduction of fungal consortia closer to practice, however, not many examples of pilot or industrial tests of artificial consortia are known [128-130]. Some of them are as follows:
- microbial consortium containing T. versicolor, P. ostreatus, Phanerochaete sp., Pseudomonas fluorecens and B. subtilis cells was applied for the treatment of non-domestic wastewater. This fungal/bacterial consortium was prepared by mixing fungal biomass pellets with suspensions of bacterial cells. The removal of colored substances (2700 Color Units 550 nm), COD (1.75 g/L) and nitrate (3 mg/L) was 91±2%, 90±4% and 17±2%, respectively, after 15 days of water treatment at a pilot plant [128];
- consortium of A. niger, Mucor hiemalis and Galactomyces geotrichum, has been tested for the treatment of real wastewater from the industry on a pilot scale station (110 L) and industrial wastewater treatment plant (1000 L). The efficiency of COD removal in industrial reactor was 50% under the influence of this consortium [129];
- consortium containing Acinetobacter oleivorans, Corynebacterium sp., Pseudomonas sp, Rhodococcus sp., Micrococcus sp. and yeast Yarrowia sp. was tested by Ecophile Co., Ltd. (Korea) in the biodegradation of hydrocarbons in soil (2300 mg/kg) contaminated with diesel fuel. This large-scale experiment involved 2 samples of 100 metric tons of contaminated soil, which were without (control) and with consortium treatment (109 cells/kg of soil). The introduction of consortium reduced pollution by 57.7% within 2 weeks, whereas in the control without the consortium, degradation was only 10.1% [130].
Thus, such positive samples of scaling use of artificial fungal consortia not only demonstrate the real-world efficacy of them but also address potential solutions encountered during practical applications.
The quality of the English is not bad, however, some expressions uncommon in the field are used, which may cause confusion to the reader. Perhaps reviewing the translation of these points would be beneficial.
We have edited the English language throughout the article.
With high respect and good wishes,
Authors of the manuscript.

Reviewer 2 Report
Comments and Suggestions for Authors
The manuscript titled "Fungi in Artificial Microbial Consortia to Solve Bioremediation Problems" is a comprehensive review that focuses on the growing interest and advancements in the creation of artificial microbial consortia, especially for various bioremediation processes.
The used literature is up-to-date and extensive. The manner of a presentation is satisfactory. Although the written review is well-written and well-structured, similar review may be already found. I suggest the addition of novel paragraphs that would enhance the practical relevance of the manuscript:
1. While the manuscript does an excellent job in reviewing the literature and discussing the theoretical aspects of using fungi in artificial microbial consortia for bioremediation, it appears to lack direct examples of practical implementation or detailed case studies. This would not only demonstrate the real-world efficacy of these consortia but also address potential challenges and solutions encountered during practical applications.
2. The good point would also be adding the discussion on the economic viability and scalability of using artificial microbial consortia in large-scale bioremediation projects. This includes considerations of cost-effectiveness, resource requirements, and logistical challenges in deploying these solutions on a larger scale.
In my opinion the adding of these two points is crucial for making this manuscript a more valuable resource for researchers and practitioners in the field of environmental biotechnology and bioremediation.
Author Response
Dear Reviewer,
We are grateful to you for the suggestions allowing us the improving of our manuscript.
Please, see our comments to your remarks and the revised text of the paper:
- While the manuscript does an excellent job in reviewing the literature and discussing the theoretical aspects of using fungi in artificial microbial consortia for bioremediation, it appears to lack direct examples of practical implementation or detailed case studies. This would not only demonstrate the real-world efficacy of these consortia but also address potential challenges and solutions encountered during practical applications.
The authors are sincerely very grateful for the Reviewer’s recommendations, as they allowed us to widen presentation of practically oriented information to the text of the review, which, it seems to us, allowed us to pay additional attention to examples of the use of artificial consortia with fungal cells and improved the review. We put this new text before the conclusion. Several new references corresponding to this text were added to the list of references. The inserted text is following:
Self-stabilization of the artificial fungi-containing consortia is sufficient for their use in non-immobilized form under laboratory conditions for the treatment of various real contaminated industrial and environmental water and soil samples [39,44,48,50,93,106,107,109]. Such studies bring the introduction of fungal consortia closer to practice, however, not many examples of pilot or industrial tests of artificial consortia are known [128-130]. Some of them are as follows:
- microbial consortium containing T. versicolor, P. ostreatus, Phanerochaete sp., Pseudomonas fluorecens and B. subtilis cells was applied for the treatment of non-domestic wastewater. This fungal/bacterial consortium was prepared by mixing fungal biomass pellets with suspensions of bacterial cells. The removal of colored substances (2700 Color Units550 nm), COD (1.75 g/L) and nitrate (3 mg/L) was 91±2%, 90±4% and 17±2%, respectively, after 15 days of water treatment at a pilot plant [128 - Céspedes-Bernal, D.N.; Mateus-Maldonado, J.F.; Rengel-Bustamante, J.A.; Quintero-Duque, M.C.; Rivera-Hoyos, C.M.; Poutou-Piñales, R.A.; Diaz-Ariza, L.A.; Castillo-Carvajal, LC.; Paez-Moralez, A.; Pedroza-Rodríguez, A.M. Non-domestic wastewater treatment with fungal/bacterial consortium followed by Chlorella sp., and thermal conversion of the generated sludge. 3 Biotech 2021, 11, 1-18. https://doi.org/10.1007/s13205-021-02780-1];
- consortium of A. niger, Mucor hiemalis and Galactomyces geotrichum, has been tested for the treatment of real wastewater from the industry on a pilot scale station (110 L) and industrial wastewater treatment plant (1000 L). The efficiency of COD removal in industrial reactor was 50% under the influence of this consortium [129 - Djelal, H.; Amrane, A. Biodegradation by bioaugmentation of dairy wastewater by fungal consortium on a bioreactor lab-scale and on a pilot-scale. J. Environ. Sci. 2013, 25, 1906-1912. https://doi.org/10.1016/S1001-0742(12)60239-3];
- consortium containing Acinetobacter oleivorans, Corynebacterium sp., Pseudomonas sp, Rhodococcus sp., Micrococcus sp. and yeast Yarrowia sp. was tested by Ecophile Co., Ltd. (Korea) in the biodegradation of hydrocarbons in soil (2300 mg/kg) contaminated with diesel fuel. This large-scale experiment involved 2 samples of 100 metric tons of contaminated soil, which were without (control) and with consortium treatment (109 cells/kg of soil). The introduction of consortium reduced pollution by 57.7% within 2 weeks, whereas in the control without the consortium, degradation was only 10.1% [130 - Lee, Y.; Jeong, S.E.; Hur, M.; Ko, S.; Jeon, C.O. Construction and evaluation of a Korean native microbial consortium for the bioremediation of diesel fuel-contaminated soil in Korea. Front. Microbiol. 2018, 9, 2594. https://doi.org/10.3389/fmicb.2018.02594].
Thus, such positive samples of scaling use of artificial fungal consortia not only demonstrate the real-world efficacy of them but also address potential solutions encountered during practical applications.
- The good point would also be adding the discussion on the economic viability and scalability of using artificial microbial consortia in large-scale bioremediation projects. This includes considerations of cost-effectiveness, resource requirements, and logistical challenges in deploying these solutions on a larger scale.
The economic viability and attractiveness of using artificial consortia containing fungi for bioremediation determines by the cost-effectiveness of cultivation of individual strains and their availability for commercial application. The availability can be satisfied by searching for and isolating the necessary fungi from natural sources or creating genetically modified strains [114,115], whereas the costs of obtaining the necessary amounts of biomass of fungal cells, according to the economic estimates just conducted, are considered low enough [114,132 - Meyer, V.; Basenko, E.; Benz, J. P.; Braus, G.; Caddick, M.; Csukai, M.; Vries, R.P.; Endy, D.; Frisvad, J.; Gunde-cimerman, N.; Haarmann, T.; Hadar, Y.; Hansen, K.; Johnson, R.; Keller, N.; Krasevec, N.; Mortensen, U.; Perez, R.; Ram, A.; Wosten, H. Growing a circular economy with fungal biotechnology: A white paper. Fungal Biol.Biotech. 2020, 7, 5. https://doi.org/10.1186/s40694-020-00095-z; Niego, A.G.T.; Lambert, C.; Mortimer, P.; Thongklang, N.;·Rapior S.; Grosse M.;·Schrey H.; Charria‑Girón E.; Walker A.; Hyde K.D.; Stadler M. The contribution of fungi to the global economy. Fungal Diversity 121, 95–137 (2023). https://doi.org/10.1007/s13225-023-00520-9]. Enzymes synthesized by fungi, as the main catalysts of many redox and hydrolytic processes, are already commercially available, although they are more expensive products compared to the fungal cells producing them. This should be noted when estimating the economic viability and scalability of using of artificial fungi-containing consortia in large-scale bioremediation projects. It is the positive aspects of the economy associated with the use of fungi in various processes, including bioremediation [133], may contribute to the wider use of artificial consortia with fungi in the near future. It is the positive aspects of the economy associated with the use of fungi in various processes, including bioremediation [133 - Bioremediation Market, By Type (In-Situ Bioremediation and Ex-Situ Bioremediation), By Services (Soil, Wastewater, and Oilfield Remediation and Others), By Technology, and By Region Forecast to 2032 (Publ. February 2023, Report ID: ER_00215); https://www.emergenresearch.com/industry-report/bioremediation-market (accessed 08 February, 2023)], may contribute to the wider use of artificial consortia with fungi in the near future.
Again, the authors are very grateful for the comments of the Reviewer given ideas about possible improvement of the manuscript.
With high respect and good wishes,
Authors of the manuscript.

Reviewer 3 Report
Comments and Suggestions for Authors
The subject of this review is of particular interest since it concerns a very current problematics in the field of bioremediation. I am pretty sure it will interest a wide range of scientists in this field. The problems I found are mostly at the level of sentence clarity and English mistakes in a very heterogenous way. Indeed, most of the review has no wording problems and the problems are concentrated to some parts such as introduction giving the sensation several authors worked on the different parts. Some remarks also concern lacking information which could be useful for the reader and improve the review. The detail is hereafter:
- Line 9: “up-to-date” looks in conflict with “growing up”. Up-to-date could be replaced by “currently growing up” for instance
- Lines 29-32: this sentence has to be rephrased; its structure is too complex to be easily understood
- L. 37-41: I do not really see why this sentence starts with however and the global sentence has to be rephased to be clear.
- L. 55-56: the sentence fragment “due to the low…conversion” is not understandable
- L. 69: only lipopeptides are cited as biosurfactant giving the sensation that fungi only produce this kind of biosurfactants whereas their chemical nature can be very diverse. So please rephrase or complete this sentence.
- L.121-123 “with the greatest…heavy metals” has to be rephrased
- L. 132: I do not understand of therefore in relation to what was said before
- L.144: it is not clear to which study “it was observed” refers
- L.332: The use OF a consortium of THE bacterium
- L.356: Do not understand to what “fungus I” refers
- L.379: destruction looks not adapted, I propose degradation
- L.386: the term purification looks not adapted; I think the authors want to say remediation
- L.397-400: I think it would be important for the reader to precise here that the PAH biodegradation pathways are described to be different in ligninolytic and non-ligninolytic fungi.
- L.555: “other pollutants” doesn’t reflect the content of this part which concerns more “pollutant mixtures”
- L.645: solving instead solve
- L.661-662: it is necessary to precise the total number used for the calculation of the percentages in the title of the figure 1 to better see the significance of these proportions
- L. 675-681: some sentences about the limitations due to the regulations concerning the spreading of GMOs are lacking here to enable the reader to have this aspect in mind
- L. 686: temperature is cited 2 times
- L. 773-774: combine and organize without -ing
Comments on the Quality of English LanguageThe quality of English has to be homogenized to have an adapted scientific English everywhere and prevent too complex sentences as detailed in the comments and suggestions for authors.
Author Response
Dear Reviewer,
We are grateful to you for the suggestions allowing us the improving of our manuscript.
Please, see our comments to your remarks and the revised text of the paper:
The problems I found are mostly at the level of sentence clarity and English mistakes in a very heterogenous way. Indeed, most of the review has no wording problems and the problems are concentrated to some parts such as introduction giving the sensation several authors worked on the different parts. Some remarks also concern lacking information which could be useful for the reader and improve the review. The detail is hereafter:
Line 9: “up-to-date” looks in conflict with “growing up”. Up-to-date could be replaced by “currently growing up” for instance
Thank you for the comment. Text has been corrected.
Lines 29-32: this sentence has to be rephrased; its structure is too complex to be easily understood
Thank you for the comment. Text has been corrected.
- 37-41: I do not really see why this sentence starts with however and the global sentence has to be rephased to be clear.
Thank you for the comment. Text has been corrected.
- 55-56: the sentence fragment “due to the low…conversion” is not understandable
Thank you for the comment. Text has been corrected.
- 69: only lipopeptides are cited as biosurfactant giving the sensation that fungi only produce this kind of biosurfactants whereas their chemical nature can be very diverse. So please rephrase or complete this sentence.
Thank you for the comment. Text has been corrected.
L.121-123 “with the greatest…heavy metals” has to be rephrased
Thank you for the comment. Text has been corrected.
- 132: I do not understand of therefore in relation to what was said before
Thank you for the comment. Text has been corrected.
L.144: it is not clear to which study “it was observed” refers
Thank you for the comment. Text has been corrected.
L.332: The use OF a consortium of THE bacterium
Thank you for the comment. Text has been corrected.
L.356: Do not understand to what “fungus I” refers
Thank you for the comment. Text has been corrected.
L.379: destruction looks not adapted, I propose degradation
Thank you for the comment. Text has been corrected.
L.386: the term purification looks not adapted; I think the authors want to say remediation
Thank you for the comment. Text has been corrected.
L.397-400: I think it would be important for the reader to precise here that the PAH biodegradation pathways are described to be different in ligninolytic and non-ligninolytic fungi.
We have added the following text to the article: “The biodegradation of PAHs by ligninolytic and non-ligninolytic fungi is characterized by two different main mechanisms due to various enzymes mainly involved in these processes. Secreted laccase, MnP, and lignin peroxidases of ligninolytic fungi are the most active catalysts acting on aromatic rings of PAHs [84 - Ghosal, D.; Ghosh, S.; Dutta, T.K.; Ahn, Y. Current state of knowledge in microbial degradation of polycyclic aromatic hydrocarbons (PAHs): a review. Front. Microbiol. 2016, 7, 1369. https://doi.org/10.3389/fmicb.2016.01369]. Oxidases can participate in the formation of free hydroxyl radicals by giving up one electron, which oxidizes PAH rings outside fungal cells. The products of such reactions are PAH-quinones and acids, whereas intracellular cytochrome P450 monooxygenases, hydroxylating low molecular weight PAHs, are the main participants in the oxidative degradation of PAHs carried out by non-ligninolytic fungi [84]. Cytochrome P450 monooxygenases oxidize PAHs to epoxides and dihydrodiols. However, in artificial microbial consortia of mixed composition, a synergistic metabolic effect can develop, leading to improved degradation of PAHs by consolidation of various catalytic mechanisms (Table 5).
The new reference was added to the text. So, numeration of references was moved.
L.555: “other pollutants” doesn’t reflect the content of this part which concerns more “pollutant mixtures”
Thank you for the comment. Text has been corrected.
L.645: solving instead solve
Thank you for the comment. Text has been corrected.
L.661-662: it is necessary to precise the total number used for the calculation of the percentages in the title of the figure 1 to better see the significance of these proportions
The total number of artificial consortia presented in the Tables 1-7 was clarified in the title of Figure 1.
- 675-681: some sentences about the limitations due to the regulations concerning the spreading of GMOs are lacking here to enable the reader to have this aspect in mind
The following additional information has been added: “Since there is no globally accepted regulatory documents concerning of genetically modified organisms spreading, in different countries the regulation of the development and release of genetically modified organisms varies depending on the purposes of their use: from a complete ban on their import, release or use to allowing their use with varying degrees of regulation. However, despite this, methods of genetic engineering of filamentous fungi continue to be actively developed and used in research around the world, making it possible to overcome many of the shortcomings of classical methods for improving strains [114,115 - Meyer, V.; Basenko, E.Y.; Benz, J.P.; Braus, G.H.; Caddick, M.X.; Csukai, M.; de Vries, R.P.; Endy, D.; Frisvad, J.C.; Gunde-Cimerman, N.; Haarmann, T.; Hadar, Y.; Hansen, K.; Johnson, R.I.; Keller, N.P.; Kraševec, N.; Mortensen, U.H.; Perez, R.; Ram, A.F.J.; Record, E.; Ross, P.; Shapaval, V.; Steiniger, C.; van den Brink, H.; van Munster, J.; Yarden, O.; Wösten, H.A.B.; Wösten, H.A. Growing a circular economy with fungal biotechnology: a white paper. Fungal Biol, Biotechnol. 2020, 7, 1-23. https://doi.org/10.1186/s40694-020-00095-z; Salazar-Cerezo, S.; de Vries, R.P.; Garrigues, S. Strategies for the development of industrial fungal producing strains. J. Fungi 2023, 9, 834. https://doi.org/10.3390/jof9080834].
- 686: temperature is cited 2 times
Thank you for the comment. Text has been corrected.
- 773-774: combine and organize without -ing
Thank you for the comment. Text has been corrected.
The quality of English has to be homogenized to have an adapted scientific English everywhere and prevent too complex sentences as detailed in the comments and suggestions for authors.
The authors are very grateful for the valuable comments and suggestions of the Reviewer enabling the quality of the article. All of them were taken into account for the introduction of specified corrections to the text. We have also edited the English language throughout the article in order to “homogenize” it and simplify all complex sentences as much as possible.
With high respect and good wishes,
Authors of the manuscript.

Round 2
Reviewer 1 Report
Comments and Suggestions for Authors
Dear Authors,
After a thorough review of your manuscript and the provided response, I would like to explain my decision to reject the article.
While I acknowledge the efforts to address the raised issues during the review, unfortunately, I did not identify substantial improvements in the article beyond the suggestions I provided. I feel that additional revisions by other researchers and new perspectives are necessary to enhance the work.
Thank you for your understanding.
Reviewer 2 Report
Comments and Suggestions for Authors
The authors have answer to all suggestions. So, in my opinion this manuscript should be published in present form.